**American Society for Microbiology | Microbiology Spectrum**

# Genomic and functional insights into aromatic compound degradation by *Delftia* strain PS-11 isolated from freshwater pufferfish

Ritu Rani Archana Kujur,[1] Sushanta Deb,[1] Tanmoy Debnath,[1] Sandesh Papade,[2] Prashant S. Phale,[2] Subrata K. Das[1]

**ABSTRACT**   This study describes the functional genomic features and aromatic compound degradation potential of *Delftia* strain PS-11, isolated from the skin mucus of a freshwater pufferfish (*Tetraodon cutcutia*). Whole-genome sequence analysis of strain PS-11 revealed genetic attributes associated with species distinctiveness. The overall genome-relatedness indices, such as *in silico* DNA-DNA hybridization, are below 70%, and average nucleotide identity and average amino acid identity were below the threshold value, that is, 95% to 96%, respectively. Phylogenomic analysis based on genome-wide core genes showed that strain PS-11 clustered with *Delftia acidovorans* NBRC 14950[T]. Furthermore, genome analysis identified multiple gene clusters involved in the degradation of aromatic compounds, including phenol, benzoic acid, and hydroxy-benzoic acid. We determined the specific activities of key metabolic enzymes in cell-free extracts, including catechol-1,2-dioxygenase, catechol-2,3-dioxygenase, protocatechuic acid 4,5-dioxygenase, and gentisate 1,2-dioxygenase. To our knowledge, the presence of *Delftia* strains associated with pufferfish has not been reported previously. These findings suggest that members of this genus occupy a broader and more diverse range of ecological habitats than previously recognized and may play a potential role in the remediation of aromatic pollutants.

**IMPORTANCE** *Delftia* strain PS-11 described in this study associated with freshwater pufferfish skin is capable of degrading aromatic compounds. Gene clusters in the form of operons involved in degradation of phenol, benzoic acid, and hydroxybenzoic acid were present. Enzymatic studies support the proposed degradation pathways. Our study suggests that the presence of xenobiotics-degrading *Delftia* strain in the skin of pufferfish adds a new ecological niche to this bacteria. The observed metabolic pathway in the degradation of aromatic compounds indicates niche-specific adaptive evolution for this bacterium.

**KEYWORDS**   pufferfish, *Delftia* strain PS-11, phylogenomic analysis, xenobiotics, biodegradation, pathway assembly

Aromatic compounds degrading microorganisms are ubiquitous in the environment, nevertheless most frequently isolated from polluted environments, rhizosphere soil, and activated sludge (1). The effect of contamination of aromatic compounds like phenol, benzoic acid, and hydroxybenzoic acid in a freshwater environment could lead to severe consequences in aquatic life. These compounds are well reported to create global problems to the aquatic and terrestrial ecosystem for their toxicity (2). Microbial degradation of aromatic compounds generally takes three basic steps: activation of the aromatic ring, ring cleavage, and breakdown of the cleavage product into the intermediates of the Krebs cycle. This process involves a set of catabolic enzymes (3),

**Peer Reviewers** Adebayo Jonathan Adeyemo, Federal University of Technology, Akure, Ondo State, Nigeria; Yvonne Marvellous Akpudo, University of Pretoria, Pretoria, South Africa; Vida Časaitė, Vilnius University, Life Sciences Center, Vilnius, Lithuania

Address correspondence to Subrata K. Das, subrata@ils.res.in.

The authors declare no conflict of interest.

depending on the genetic characteristics of bacteria. Until now, numerous studies have characterized the degradation pathways for aromatic organic compounds (4). Furthermore, bacteria isolated from a natural ecosystem have shown biochemical, genetic, and physiological evidence for degradation of aromatic compounds present as hazardous environmental pollutants (5).

Several studies have demonstrated that bacteria can utilize aromatic compounds as a carbon source, involving specific metabolic pathways under either aerobic or anaerobic conditions (6–9). Researchers employ several methods, such as measuring whole-cell oxygen consumption with probable intermediates, determining specific enzyme activities in cell-free extracts from cells grown on these compounds, and evaluating the bacterial growth on various probable pathway intermediates (10). However, limited information is available regarding the specific mechanisms and pathways utilized for the degradation of phenol, benzoic acid, and hydroxybenzoic acid by a *Delftia* species.

*Delftia* are motile, aerobic, non-fermenting, gram-negative rods that belong to phylum *Pseudomonadota* and class of *Betaproteobacteria*. Majority of studies with *Delftia* have focused on phylogenomic and taxonomic characterization of individual species. Strains have been isolated from diverse niches that indicate the metabolic capabilities and saprophytic lifestyle of *Delftia* (11, 12). Earlier studies have demonstrated habitat plasticity of this group of bacteria isolated from a wide range of ecological niches. *Delftia tsuruhatensis*, initially isolated from sludge, have been found in the healthy adult zebrafish (*Danio rerio*) skin mucus community, water systems, in fish guts (13–15), and from the injured eye of freshwater Nile tilapia (*Oreochromis niloticus*) (16). Interestingly, *Delftia tsuruhatensis* BM90, isolated from the Tyrrhenian Sea, shows its ability to degrade or metabolize a wide array of phenolic compounds and other organic pollutants (17–20). Therefore, *Delftia* spp. have attracted attention as a promising resource for its bioremediation applications.

This study reports on a bacterium, *Delftia* strain PS-11, isolated from the skin mucus of freshwater pufferfish (*Tetraodon cutcutia*). The presence of *Delftia* in the freshwater pufferfish is not known so far. Phylogenetic analysis and the overall genome related indices (OGRI) supported the novelty of the strain PS-11. In conclusion, we described the metabolic pathways and mapped the genes responsible for the phenol, benzoic acid, and hydroxybenzoic acid degradation in *Delftia* strain PS-11.

## RESULTS AND DISCUSSION

### Phenotype of strain PS-11

The cells of strain PS-11 were gram-negative, rod-shaped, and positive for oxidase and catalase. Obligate aerobe. Strain PS-11 metabolizes fructose, sodium gluconate, glycerol, salicin, inositol, mannitol, arabitol, erythritol, rhamnose, cellobiose, melezitose, α-methyl-D-mannoside, xylitol, *o*-nitrophenyl β-galactoside, D-arabinose, and citrate utilization. Strain failed to assimilate lactose, xylose, galactose, raffinose, sucrose, L-arabinose, mannose, dulcitol, sorbitol, adonitol, sorbose, malonate, α-methyl-D-glucoside, and esculin hydrolysis. The phenotypic features and the assimilation of various carbon sources resemble the properties of the type species of *Delftia*.

### Genome features of *Delftia* strain PS-11

The genome size of strain PS-11 was found to be 5.5 Mb with an average guanine-plus-cytosine (G+C) content of 65.5%. A total of 47,061,338 base pair reads were obtained through whole-genome sequencing. The lengths of the forward and reverse reads were 23,450,336 bp each. Sequence alignment against the reference genome resulted in 99.75% of sequences mapped, of which 98.09% were correctly oriented and placed as paired ends. The hybrid genome assembly resulted in a continuous, non-fragmented single contig spanning 5,538,489 bp, achieving an N50 value equal to that length and covering 81.32% of the genome. The genome comprises 6,798 coding sequences, genes for 87 transfer RNAs, 6 5S ribosomal RNAs (rRNAs), 6 16S rRNA, and 6 23S rRNA were annotated (GenBank accession number: JACSYA000000000).

## Overall genome-related indices and phylogenetic analysis

Whole-genome sequencing is regarded as a promising tool for taxonomic and phylogenetic analysis of microorganisms. In this regard, the overall genome relatedness index (OGRI) was evaluated to describe *Delftia* strain PS-11 as a novel species of the genus *Delftia*. The average nucleotide identity (ANI) and average amino acid identity (AAI) values of strain PS-11 with the reference genome for the validly named type species available in the National Center for Biotechnology Information (NCBI) database were below the threshold values (95%–96%), justifying for bacterial species delineation (21). Furthermore, *in silico* genomic DNA similarity values were lower than the 70% cut-off to define bacterial species (22). The ANI, AAI, and *in silico* DNA–DNA hybridization (isDDH) data indicate that the strain PS-11 can be proposed as a novel species in the genus *Delftia* (Table 1). The essential component of the current taxonomy is the genome-wide comparative phylogenomic analysis based on core genes superior to those found on single-gene markers, stating that core gene-based phylogenomic analysis showed that strain PS-11 is close to *Delftia acidovorans* NBRC 14950$^T$. However, based on OGRI (AAI, ANI, and isDDH), strain PS-11 is proposed to be a novel species of the genus *Delftia* (Fig. 1).

## Comparative genomics

Comparative functional profiling revealed that *Delftia* strain PS-11 possesses distinct functional attributes as evidenced by its separation from the other *Delftia* species in the functional clustering analyses. Considerable heterogeneity among the species was observed, as representing less than 40% BLAST similarity (see Fig. S1 at https://figshare.com/s/479569d8fb80f976b8ea). Clusters of Orthologous Genes (COG) functional profiling has demonstrated that *Delftia* strain PS-11 genome exhibited relatively higher abundance of genes encoding for two COG functional categories, that is, signal transduction mechanisms (T) and translation, ribosomal structure, and biogenesis (J) (Fig. 2). This pattern is consistent with previous studies on bacteria capable of degrading aromatic compounds (23, 24). Additionally, genes involved in biodegradation pathways were selectively obtained from Kyoto Encyclopedia of Genes and Genomes (KEGG) analysis, and genes encoding nine KEGG xenobiotic biodegradation pathways (naphthalene, toluene, fluorobenzoate, xylene, chlorocyclohexane and chlorobenzene, styrene, dioxin, and benzoic acid) were enriched in *Delftia* strain PS 11 genome (Fig. 3). Researchers previously employed a similar KEGG-based approach to analyze the xenobiotic degradation capabilities among different bacterial members (25, 26). To provide a comprehensive overview, total genes encoding metabolic potential and degradation capability of four *Delftia* species were listed along with their KEGG function (see Table S1 at https://figshare.com/s/479569d8fb80f976b8ea). Furthermore, comparative analysis of genes involved in xenobiotics degradation of *Delftia* PS-11 with other strains based on KEGG pathway was listed in (Table 2). The combination of COG and KEGG functional analysis together supports the notion of biodegradation capability of *Delftia* PS11 strain isolated from freshwater pufferfish skin. A pan-genome analysis of *Delftia* strain

**TABLE 1** Comparison of the genomic characteristics of strain *Delftia* strain PS-11 with closely related species of *Delftia*

| Sl. no. | Strains | Accession no. | Size (Mb) | 16S rRNA similarity (%) | ANI (%) | AAI (%) | isDDH (%) |
|---------|---------|---------------|-----------|-------------------------|---------|---------|-----------|
| 1 | *Delftia* strain PS-11 MTCC 13821 | JACSYA000000000 | 5.53 | 100 | 100 | 100 | 100 |
| 2 | *Delftia acidovorans* NBRC 14950$^T$ | BCZP00000000 | 6.6 | 98.69 | 85.45 | 85.93 | 29.70 |
| 3 | *Delftia lacustris* LMG 24775$^T$ | FNPE00000000 | 7.3 | 98.7 | 85.53 | 86.05 | 29.70 |
| 4 | *Delftia tsuruhatensis* NBRC 16741$^T$ | BCTO00000000 | 6.6 | 98.69 | 85.48 | 85.86 | 29.70 |

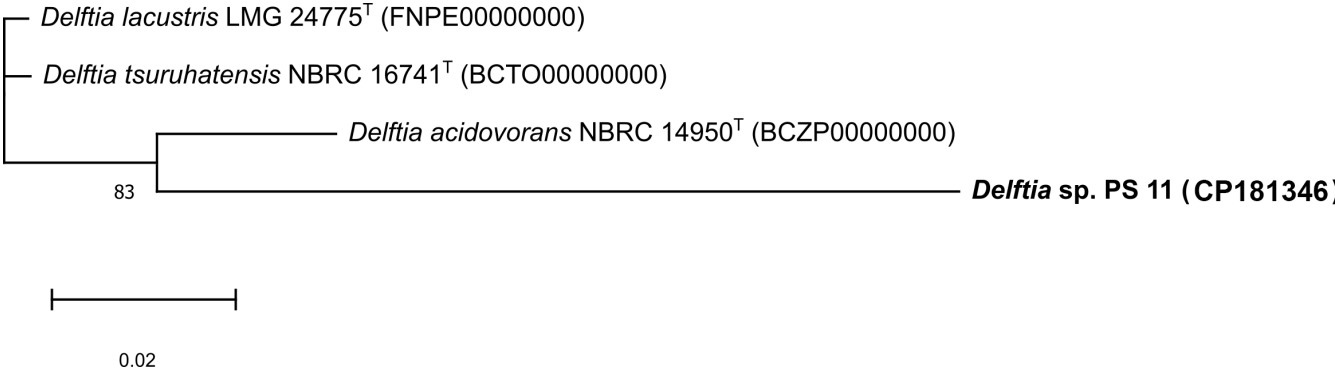

FIG 1 Phylogenetic tree based on the alignment of core genes of *Delftia* strain PS-11.

PS-11 and its phylogenetically close relatives, that is, *Delftia acidovorans* NBRC 14950[T], *Delftia lacustris* LMG 24775[T], and *Delftia tsuruhatensis* NBRC 16741[T], shared 670 core genes across all *Delftia* genomes, accordingly proposed as a single copy gene cluster (see Fig. S1 at https://figshare.com/s/479569d8fb80f976b8ea). The pan-genome analysis of 4 genomes comprised 12,602 Shell genes and 13,272 total genes, of which 24, 56, and 59 were exclusively present in *Delftia tsuruhatensis* NBRC 16741[T], *Delftia* sp. PS-11, and *Delftia lacustris* LMG 24775[T], respectively (see Fig. S1 at https://figshare.com/s/479569d8fb80f976b8ea). The presence of a unique gene pool specific to the *Delftia* strain PS-11 genome may confer its enhanced capability for aromatic compound biodegradation.

Furthermore, genome-wide genes connected to subsystems and their distribution in different functional categories for the strain PS-11 and *Delftia acidovorans* NBRC 14950[T] were determined. As a result, 5,134 protein-coding genes from strain PS-11 and 6,214 from *Delftia acidovorans* NBRC 14950[T] were predicted. Among these, 1,457 genes of strain PS-11 and 1,498 of *Delftia acidovorans* NBRC 14950[T] could be annotated by Rapid Annotation of microbial genomes using Subsystems Technology (RAST)'s automated homology analysis procedure and assigned functional categories (see Fig. S2 at https://figshare.com/s/479569d8fb80f976b8ea). Unique gene features of *Delftia* sp. PS-11 involved in the metabolism of aromatic compounds corresponding to distinct subsystems with closely related type strain *Delftia acidovorans* NBRC 14950[T] are catechol branch of *beta*-ketoadipate pathway, central *meta*-cleavage pathway of aromatic compound degradation, benzoic acid degradation, and aromatic dioxygenase (see Table S1 at https://figshare.com/s/479569d8fb80f976b8ea).

## Ability to degrade range of aromatics and metabolic diversity

Strain PS-11 metabolizes various aromatic compounds including phenol, benzoic acid, salicylic acid, 3-hydroxybenzoic acid (3-HBA), and 4-hydroxybenzoic acid (4-HBA) as the sole source of carbon and energy. Metabolic and genomic versatility of genus *Delftia* is well reported, with the ability to metabolize a diverse array of aromatic compounds including aniline, naphthalene, phenanthrene, and others (27, 28). Aromatic compounds are commonly metabolized by microbes through intermediates like benzoic acid or various hydroxybenzoic acids. These intermediates are further processed into central metabolic compounds such as catechol, protocatechuic acid, and gentisic acid. While bacterial breakdown of benzoic acid and salicylic acid is well studied (29, 30), the ability of a single bacterial strain to aerobically degrade phenol, benzoic acid, salicylic acid, and 3-/4-HBA has rarely been documented (31, 32). Strain PS-11 could metabolize benzoic acid and all three hydroxybenzoic acid isomers as the sole source of carbon and energy, reaching optical density ($OD_{540nm}$) up to 0.2–0.3 after 36–48 h of growth.

Cell-free extract (CFE) prepared from strain PS-11 cells grown on benzoate and phenol showed specific activity (nmol min$^{-1}$ mg$^{-1}$ protein) of catechol 1,2-dioxygenase (178 ±

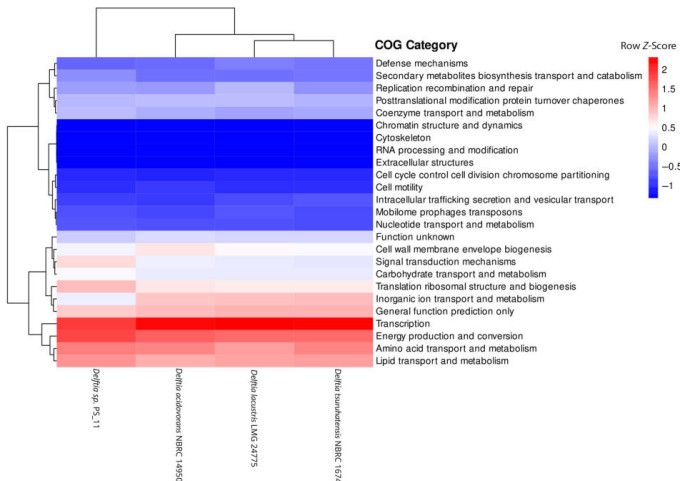

**FIG 2** Heatmap depicting differentially enriched Clusters of Orthologous Genes (COG) metabolic functions across *Delftia* genomes. Raw *z*-score (scale range from −4 to +4) indicates abundance of genes associated with each functional category illustrated with blue (low abundance) and red (high abundance) colors. The hierarchical clustering of genomes was performed using a weighted Bray-Curtis approach.

61) and catechol 2,3-dioxygenase (749 ± 91), respectively (Table 3). The CFE prepared from salicylic acid and 3-HBA grown cells showed presence of gentisate 1,2-dioxygenase activity (124 ± 16 and 547 ± 74), respectively. Time-dependent spectral changes in the presence of CFE of strain PS-11 grown on 4-HBA showed a decrease in absorbance at 290 nm (λmax for protocatechuate) and an increase in absorbance at 410 nm, indicating the presence of protocatechuate 4,5-dioxygenase activity (107 ± 28) (Table 3). These suggest that strain PS-11 possesses metabolic (lower/downstream) pathways for degradation of all three central intermediates (i.e., catechol, protocatechuic acid, and gentisic acid) in aerobic degradation of aromatics, highlighting metabolic diversity. Moreover, the ability of strain PS-11 to degrade phenol as well as other aromatic compounds indicates its potential in environmental restoration, particularly in water bodies where aromatic contamination is prevalent due to industrial and domestic waste dumping/discharge. Based on the growth and enzymatic activity, the degradation pathways for the aromatic compounds by the strain *Delftia* PS-11 have been proposed (Fig. 4).

## Functional genome-mining for aromatic compound degradation pathways in strain PS-11

The industrial revolution has led to the introduction of new pollutants in the environment, which also ushered the evolution of new catabolic pathways (33). The absence or withdrawal of such selective pressures often leads to the loss of the catabolic property, if it is plasmid-mediated, and even in cases of genome organization, where it is associated with mobile genetic elements that can move within or between genomes and modify and change the overall structure of a genome by inserting genomic islands. These features play a significant role in the evolution of microbial community, where these evolved microbes can be ideal candidates for effective bioremediation either alone or in consortium (34, 35). The *in silico* analysis of the strain PS-11 genome revealed the genes coding for enzymes involved in the metabolic pathways which are biochemically characterized from PS-11 (Fig. 4) and their arrangement on the genome (Fig. 5).

## Phenol degradation pathway

Strain PS-11 utilizes phenol as the sole source of carbon and energy via ring hydroxylation pathway (Fig. 4). Phenol catabolism is encoded within a compact operon comprising regulatory, structural, and accessory genes arranged in a functionally coordinated cluster (Fig. 5A). The operon begins with *dmpR*, a σ⁵⁴-dependent transcriptional activator that

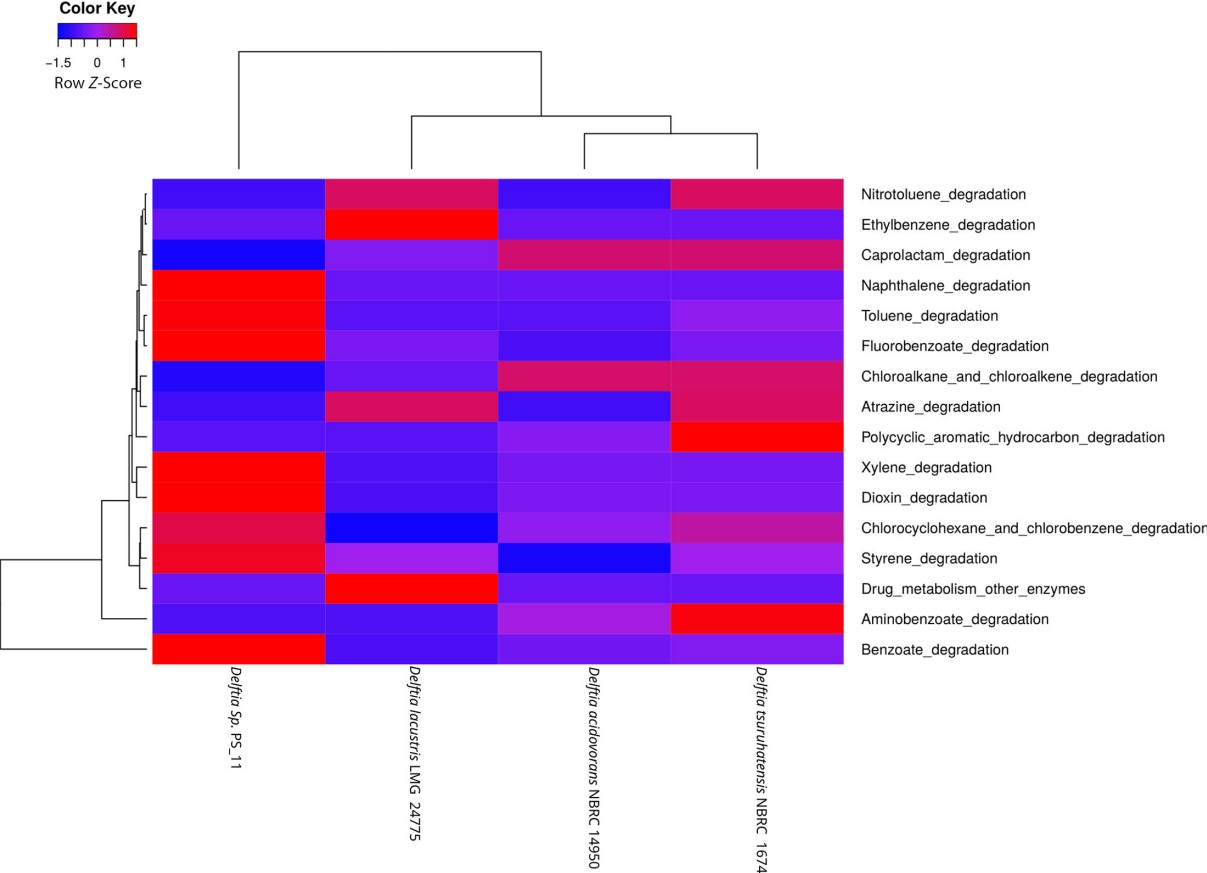

**FIG 3** Heatmap depicting differentially enriched Kyoto Encyclopedia of Genes and Genomes xenobiotic biodegradation pathways across *Delftia* genomes. Raw z-score (scale range from −4 to +4) indicates abundance of genes associated with each functional category illustrated with blue (low abundance) and red (high abundance) colors. The hierarchical clustering of genomes was performed using a weighted Bray-Curtis approach.

senses phenolic effectors and activates expression of the downstream phenol hydroxy-lase genes. The presence of *dmpR* highlights the regulatory similarity of PS-11 with *Pseudomonas* sp. CF600, the archetypal model for phenol degradation (36).

The structural genes encode a multicomponent phenol hydroxylase system (*dmpL*, *dmpN*, *dmpO*, and *dmpP*), which catalyzes the initial hydroxylation of phenol to catechol. Specifically, *DmpL* (P1 oxygenase), *DmpN* (P3 oxygenase), and *DmpO* (P4 oxygenase) constitute the oxygenase subunits, while *DmpP* functions as the FAD- and [2Fe-2S]-containing reductase, supplying electrons to the oxygenase complex. An additional iron-sulfur protein encoded within the operon likely participates in electron transfer, enhancing catalytic efficiency under variable redox conditions. The catechol intermediate generated is subsequently cleaved by catechol 2,3-dioxygenase (C23DO; EC 1.13.11.2), channeling carbon flow into the meta-cleavage pathway, which produces pyruvic acid and acetaldehyde for entry into central metabolism. This pathway enables growth on phenol as the sole carbon source and has been well documented in pseudomonads (37, 38).

Notably, the PS-11 operon also contains multiple hypothetical proteins interspersed between structural genes. While their functions remain uncharacterized, similar accessory ORFs have been observed in other phenol-degrading clusters, where they may contribute to operon stability, fine-tuned regulation, or stress response (36). Comparative insights suggest that while phenol hydroxylase operons in *Pseudomonas* spp. (e.g., *Pseudomonas* sp. CF600) and *Acinetobacter* strains often reside on plasmids (39), in *Delftia* strain PS-11, this operon is chromosomally encoded, possibly reflecting stable

**TABLE 2** Comparative analysis of genes involved in xenobiotics degradation of *Delftia* PS-11 with other strains based on KEGG pathway analysis[a]

| KEGG xenobiotic degradation pathway/KEGG symbol | 1 | 2 | 3 | 4 | KEGG protein name/KEGG orthologous genes | Reference gene/ UniProt accession |
|---|---|---|---|---|---|---|
| Benzoate | | | | | | |
| benA-xylX | + | − | − | − | Benzoate/toluate 1,2-dioxygenase subunit alpha/K05549 | A0A023WTD0 |
| benD-xylL | + | − | − | − | Dihydroxycyclohexadiene carboxylate dehydrogenase/K05783 | A0A023WUH4 |
| catA | + | − | + | + | Catechol 1,2-dioxygenase/K03381 | A0A010RM58 |
| catB | + | + | + | + | Muconate cycloisomerase/K01856 | A0A023WTJ3 |
| catC | + | + | + | + | Muconolactone D-isomerase/K03464 | A0A023WTC5 |
| pcaD | + | + | + | + | 3-oxoadipate enol-lactonase/K01055 | A0A023WVW3 |
| pcaI | + | + | + | − | 3-oxoadipate CoA-transferase, alpha subunit/K01031 | A0A024HMG1 |
| pcaF | + | + | + | + | 3-oxoadipyl-CoA thiolase/K07823 | A0A023WV56 |
| fadA , fadI | + | + | + | + | Acetyl-CoA acyltransferase/K00632 | A0A023WQU9 |
| dmpB, xylE | + | − | − | − | Catechol 2,3-dioxygenase/K00446 | A0A076MT39 |
| dmpH, xylII, nahK | + | − | − | − | 2-oxo-3-hexenedioate decarboxylase/K01617 | A0A024HK95 |
| mhpD | + | − | − | − | 2-keto-4-pentenoate hydratase/K02554 | A0A024HKG8 |
| mhpE | + | − | − | − | 4-hydroxy 2-oxovalerate aldolase/K01666 | A0A024HDK1 |
| mhpF | + | − | − | − | Acetaldehyde dehydrogenase/K04073 | A0A024HJ81 |
| badA | + | − | + | + | Benzoate-CoA ligase/K04110 | A0A060NHY8 |
| nagX | + | + | + | + | 3-hydroxybenzoate 6-monooxygenase/K22270 | A0A059MTD0 |
| praC, xylH | + | − | − | − | 4-oxalocrotonate tautomerase/K01821 | A0A010SCR7 |
| badA | + | + | − | − | Benzoate-CoA ligase/K04110 | A0A060NHY8 |
| pimC | + | + | + | + | Pimeloyl-CoA dehydrogenase large subunit/K04118 | A0A023WY93 |
| boxA | + | + | + | + | Benzoyl-CoA 2,3-epoxidase subunit A/K15511 | A0A060NHZ0 |
| boxC | + | + | + | + | Benzoyl-CoA-dihydrodiol lyase/K15513 | A0A060NLQ0 |
| fadJ | + | + | + | + | 3-hydroxyacyl-CoA dehydrogenase/K01782 | A0A024EHL2 |
| GCDH, gcdH | + | + | + | + | Glutaryl-CoA dehydrogenase/K00252 | A0A010R256 |
| paaF, echA | + | + | + | − | Enoyl-CoA hydratase/K01692 | A0A011UN55 |
| paaH, hbd, fadB | + | + | + | + | 3-hydroxybutyryl-CoA dehydrogenase/K00074 | A0A010QZK5 |
| ACAT, atoB | + | + | + | + | Acetyl-CoA C-acetyltransferase/K00626 | A0A010R8Q8 |
| 4-Hydroxybenzoate | | | | | | |
| pobA | + | + | + | + | p-hydroxybenzoate 3-monooxygenase/K00481 | A0A023X2Q4 |
| ligA | + | + | − | + | Protocatechuate 4,5-dioxygenase, alpha chain/K04100 | A0A060DXX8 |
| ligC | + | + | + | + | 2-hydroxy-4-carboxymuconate semialdehyde hemiacetal dehydrogenase/K10219 | A0A060DXY2 |
| ligK | + | + | + | + | 4-hydroxy-4-methyl-2-oxoglutarate aldolase/K10218 | A0A011TDK3 |
| pcaB | + | + | + | + | 3-carboxy-cis,cis-muconate cycloisomerase/K01857 | A0A023X294 |
| pcaC | + | + | + | + | 4-carboxymuconolactone decarboxylase/K01607 | A0A010S689 |
| galD | + | + | + | + | 4-oxalomesaconate tautomerase/K16514 | A0A060DP97 |
| ligJ | + | + | + | + | 4-oxalmesaconate hydratase/K10220 | A0A031FTK6 |
| ybgC | + | + | + | + | 4-hydroxybenzoyl-CoA thioesterase/K01075 | A0A023WQH1 |
| chcAa | − | + | + | + | Cyclohexane-1-carboxylate 4-trans-hydroxylase/K27672 | A0A0P0RNW6 |
| Aminobenzoate | | | | | | |
| mdlC | + | + | + | + | Benzoylformate decarboxylase/K01576 | A0A023X1R2 |
| amiE | + | + | + | + | Amidase/K01426 | A0A010Q136 |
| badA | + | + | + | − | Benzoate-CoA ligase/K04110 | A0A060NHY8 |
| abmG | + | + | + | + | 2-aminobenzoate-CoA ligase/K08295 | A0A060DI71 |
| abmA | − | + | + | + | Anthraniloyl-CoA monooxygenase/K09461 | A0A068QR52 |
| vanA | + | + | + | + | Vanillate monooxygenase/K03862 | A0A060NL68 |
| ligA | + | + | − | + | Protocatechuate 4,5-dioxygenase, alpha chain/K04100 | A0A060DXX8 |
| ligC | + | + | + | + | 2-hydroxy-4-carboxymuconate semialdehyde hemiacetal dehydrogenase/K10219 | A0A060DXY2 |
| desB, galA | + | + | + | + | Gallate dioxygenase/K04099 | A0A076PSU9 |
| paaF, echA | + | + | + | + | Enoyl-CoA hydratase/K01692 | A0AK0011UN55 |

(*Continued on next page*)

**TABLE 2** Comparative analysis of genes involved in xenobiotics degradation of *Delftia* PS-11 with other strains based on KEGG pathway analysis*a* (*Continued*)

| KEGG xenobiotic degradation pathway/KEGG symbol | 1 | 2 | 3 | 4 | KEGG protein name/KEGG orthologous genes | Reference gene/ UniProt accession |
|---|---|---|---|---|---|---|
| ethA | − | − | − | + | Monooxygenase/K10215 | A0A024HM19 |
| antA | − | − | − | + | Anthranilate 1,2-dioxygenase/K05599 | A0A024HFQ7 |
| Chlorocyclohexane and chlorobenzene | | | | | | |
| dmpK, poxA, tomA0 | + | − | − | − | Phenol/toluene 2-monooxygenase/K16249 | A0A024HES8 |
| CMBL | + | + | + | + | Carboxymethylenebutenolidase/K01061 | A0A010QKB9 |
| catA | + | − | + | + | Catechol 1,2-dioxygenase/K03381 | A0A010RM58 |
| catB | + | + | + | + | Muconate cycloisomerase/K01856 | A0A023WTJ3 |
| dmpB, xylE | + | − | − | − | Catechol 2,3-dioxygenase/K00446 | A0A076MT39 |
| dehH | + | + | + | + | Haloacetate dehalogenase/K01561 | A0A024E5A3 |
| Toluene | | | | | | |
| dmpK, poxA, tomA0 | + | − | − | − | Phenol/toluene 2-monooxygenase/K16249 | A0A024HES8 |
| xylB | + | − | + | − | Aryl-alcohol dehydrogenase/K00055 | A0A023X739 |
| catA | + | − | + | + | Catechol 1,2-dioxygenase/K03381 | A0A010RM58 |
| catB | + | + | − | + | Muconate cycloisomerase/K01856 | A0A023WTJ3 |
| CMBL | + | + | + | + | Carboxymethylenebutenolidase/K01061 | A0A010QKB9 |
| Xylene | | | | | | |
| xylB | + | − | − | − | Aryl-alcohol dehydrogenase/K00055 | A0A023X739 |
| benA-xylX | + | − | − | − | Benzoate/toluate 1,2-dioxygenase subunit alpha/K05549 | A0A023WTD0 |
| benD-xylL | + | − | − | − | Dihydroxycyclohexadiene carboxylate dehydrogenase/ K05783 | A0A023WUH4 |
| dmpB, xylE | + | − | − | − | Catechol 2,3-dioxygenase/K00446 | A0A076MT39 |
| praC, xylH | + | − | − | − | 4-oxalocrotonate tautomerase/K01821 | A0A010SCR7 |
| dmpH, xylI, nahK | + | − | − | − | 2-oxo-3-hexenedioate decarboxylase/K01617 | A0A024HK95 |
| mhpD | + | − | − | − | 2-keto-4-pentenoate hydratase/K02554 | A0A024HKG8 |
| mhpE | + | − | − | − | 4-hydroxy 2-oxovalerate aldolase/K01666 | A0A024HDK1 |
| mhpF | + | − | − | − | Acetaldehyde dehydrogenase/K04073 | A0A024HJ81 |
| Dioxin | | | | | | |
| praC, xylH | + | − | − | − | 4-oxalocrotonate tautomerase/K01821 | A0A023WYQ5 |
| dmpH, xylI, nahK | + | − | − | − | 2-oxo-3-hexenedioate decarboxylase/K01617 | A0A024HK95 |
| mhpD | + | − | − | − | 2-keto-4-pentenoate hydratase/K02554 | A0A024HKG8 |
| mhpE | + | − | − | − | 4-hydroxy 2-oxovalerate aldolase/K01666 | A0A024HDK1 |
| mhpF | + | − | − | − | Acetaldehyde dehydrogenase/K04073 | A0A024HJ81 |
| Styrene | | | | | | |
| amiE | + | + | + | + | Amidase/K01426 | A0A010Q136 |
| feaB, tynC | + | + | + | + | Phenylacetaldehyde dehydrogenase/K00146 | A0A023X4N7 |
| maiA, GSTZ1 | + | + | + | + | Maleylacetoacetate isomerase/K01800 | A0A010RFJ0 |
| FAH, fahA | + | + | + | + | Fumarylacetoacetase/K01555 | A0A010QGK9 |
| dmpB, xylE | + | − | − | − | Catechol 2,3-dioxygenase/K00446 | A0A076MT39 |
| pct | + | − | + | + | propionate CoA-transferase/K01026 | A0A060HZB4 |

*a*1, *Delftia* strain PS-11 MTCC 13821; 2, *Delftia acidovorans* NBRC 14950[T]; 3, *Delftia lacustris* LMG 24775[T]; 4, *Delftia tsuruhatensis* NBRC 16741[T]. +, present; −, absent.

genomic integration for adaptation to soil or contaminated environments. The overall gene organization (Fig. 5A) with regulator (*dmpR*) placed upstream of hydroxylase genes and C23DO positioned downstream is highly conserved, underscoring strong selective pressure to maintain an efficient and tightly regulated phenol utilization system.

## Benzoic acid degradation pathway

In strain PS-11, the benzoic acid degradation genes are organized as a contiguous operon consisting of nine genes (Fig. 5B). The cluster begins with a benzoic acid MFS transporter (*benK*), which facilitates substrate uptake, followed by 1,2-dihydroxy-cyclohexa-3,5-diene-1-carboxylate dehydrogenase (EC 1.3.1.25), an enzyme catalyzing the conversion of the dihydrodiol intermediate formed during benzoic acid ring

**TABLE 3** Specific activities of different enzymes involved in the degradation of phenol, benzoic acid, and hydroxybenzoic acid by *Delftia* strain PS-11

| Enzymes[b] | Specific activity (nmol min$^{-1}$ mg$^{-1}$ protein) in cell-free extract of *Delftia* strain PS-11 cells grown on | | | | |
|---|---|---|---|---|---|
| | Phenol | Benzoic acid | 2-Hydroxybenzoate (salicylic acid) | 3-Hydroxybenzoic acid | 4-Hydroxybenzoic acid |
| Catechol 1,2-dioxygenase | ND[a] | 178 ± 61 | ND | ND | ND |
| Catechol 2,3-dioxygenase | 749 ± 91 | ND | 10 ± 0.7 | ND | ND |
| Protocatechuate 3,4-dioxygenase | ND | ND | ND | ND | ND |
| Protocatechuate 4,5-dioxygenase | ND | ND | ND | ND | 107 ± 28 |
| Gentisate 1,2-dioxygenase | ND | ND | 124 ± 16 | 547 ± 74 | ND |

[a]ND, not detected.
[b]Enzyme activity assays were performed in triplicate using at least three independent biological replicates, and results are expressed as mean ± SD.

hydroxylation to catechol precursors. The central catabolic machinery is encoded by the *benABCD* genes, which together constitute the benzoate 1,2-dioxygenase system. This multicomponent oxygenase initiates degradation by catalyzing the hydroxylation of the aromatic ring, yielding catechol as a major metabolic intermediate (40). Catechol is further metabolized through the ortho-cleavage branch of the *β*-ketoadipate pathway, a conserved route for aromatic compound mineralization. In PS-11, catechol 1,2-dioxygenase (*catA*; EC 1.13.11.1) cleaves catechol into *cis,cis*-muconic acid, which is subsequently cyclized by muconate cycloisomerase (*catB*; EC 5.5.1.1). Transcriptional control is mediated by the LysR-type regulator *CatR*, which responds to *cis,cis*-muconic acid and regulates the expression of catabolic genes (41).

A notable feature of the PS-11 operon is the presence of RpoH, a sigma factor typically associated with stress responses. Its integration within the catabolic operon suggests a possible coupling of aromatic degradation with environmental adaptation, potentially enhancing survival in polluted or fluctuating niches. It is established that the *rpoH* gene in *Escherichia coli* produces the sigma 32 (σ$^{32}$) protein, a key factor that directs RNA polymerase to transcribe essential heat shock genes, effectively shielding cells from protein damage caused by sudden temperature rises or environmental stressors (42). Integration is unusual in *Pseudomonas* spp., where regulatory genes such as *catR* are typically located externally to the benzoate dioxygenase gene cluster (40). The compact architecture in PS-11, where transporter, catabolic, and regulatory genes are co-located, resembles horizontally acquired catabolic modules and points to a streamlined strategy for benzoic acid utilization (43). Comparative genomics reveals that other *Delftia* strains also encode benzoic acid and aromatic catabolic functions, but their genomic organization differs. For instance, *D. acidovorans* Cs1-4 carries benzoic acid degradation genes embedded within the *phn* genomic island (28), while *D. tsuruhatensis* ULwDis3 contains multiple aromatic catabolic operons, including an incomplete benzoic acid pathway (27).

## Hydroxybenzoic acid catabolism pathways

Genome analysis revealed multiple operons dedicated to the catabolism of hydroxybenzoic acid in strain PS-11, which funnel into two distinct central pathways: the gentisic acid and protocatechuic acid branches. A putative salicylic acid operon was identified (Fig. 5C), encoding an MFS-type transporter, a putative n-hydroxybenzoic acid hydroxylase, and accessory proteins including an H-NS–like DNA-binding protein and hemolysin-like CBS proteins. Although atypical in gene composition compared to the canonical *nahG*-mediated salicylate pathway in *Pseudomonas* spp. (44), this cluster likely mediates the conversion of salicylic acid to catecholic intermediates that converge on the gentisic acid route. For 3-HBA, PS-11 harbors a complete gentisic acid pathway operon (Fig. 5D) comprising genes for 3-hydroxybenzoic acid 6-monooxygenase, gentisate 1,2-dioxygenase, fumarylpyruvate hydrolase, and maleylpyruvate isomerase, under the control of a MarR-family regulator. This arrangement resembles the gentisic acid (*nag*) operons of *Ralstonia eutropha* and *Pseudomonas alcaligenes* (45, 46) and establishes that salicylic acid and 3-HBA are metabolized via the gentisic acid cleavage pathway in PS-11 (Fig. 4).

In contrast, 4-HBA metabolism proceeds via protocatechuic acid (Fig. 4). Two distinct *pobA*-containing operons were identified: one organized with LysR- and IclR-type

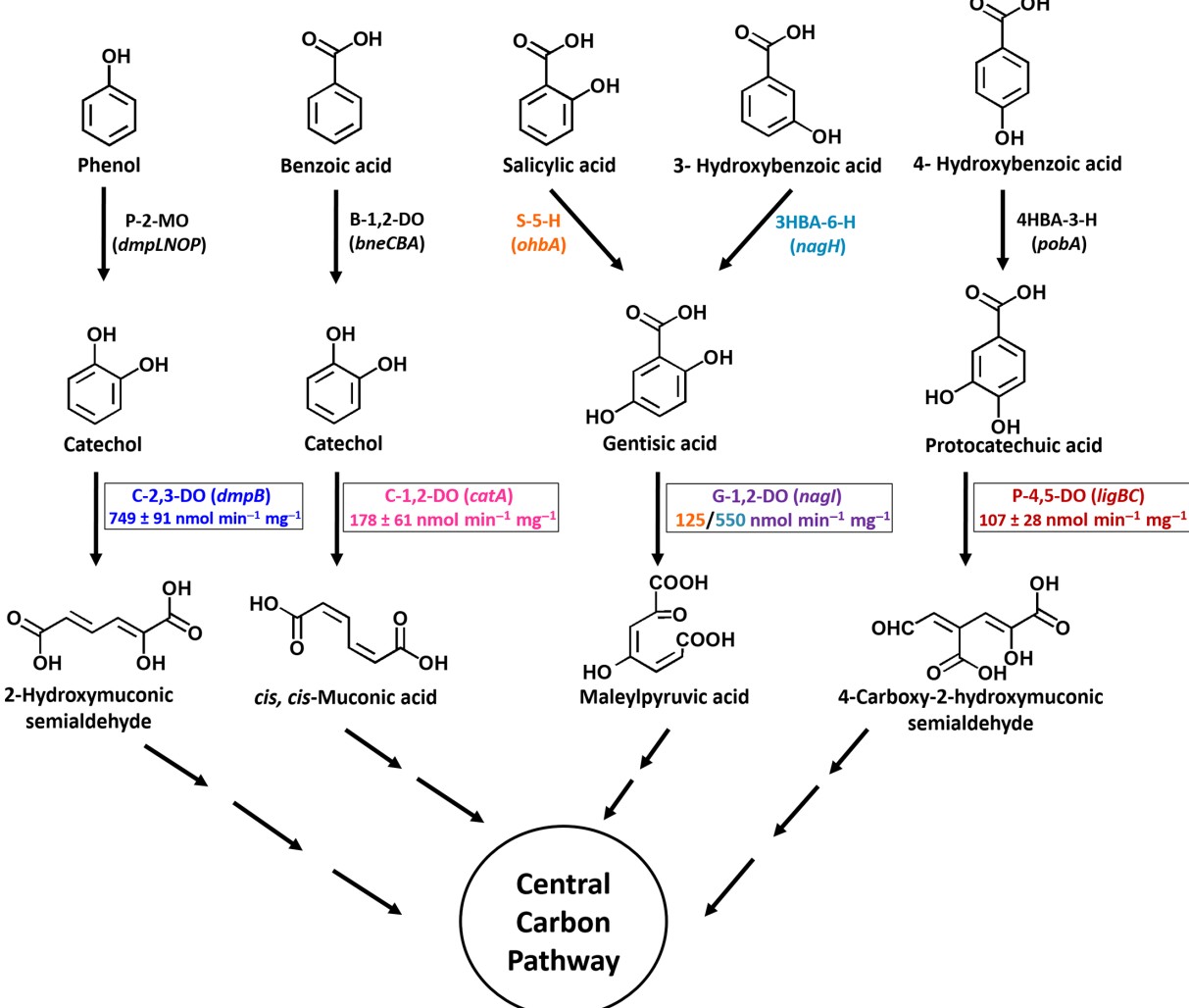

**FIG 4** Proposed pathways for degradation of phenol, benzoic acid, salicylic acid, 3-HBA, and 4-HBA by *Delftia* sp. strain PS-11. Pathways are proposed on the basis of genome analysis, growth, and enzyme activity analysis. Enzymes involved are: P-2-MO, phenol 2-monooxygenase; B-1,2-DO, benzoic acid 1,2-dioxygenase; S-5-H, salicylic acid 5-hydroxylase; 3HBA-6-H, 3-hydroxybenzoic acid 6-hydroxylase; 4HBA-3-H, 4-hydroxybenzoic acid 3-hydroxylase; C-1,2-DO, catechol 1,2-dioxygenase; C-2,3-DO, catechol 2,3-dioxygenase; G-1,2-DO, gentisate-1,2-dioxygenase; P-4,5-DO, protocatechuate-4,5-dioxygenase.

regulators and a second canonical *pobR-pobA* module (Fig. 5E). This dual organization is unusual, as most bacteria harbor a single *pobA* gene adjacent to its regulator (e.g., *Pseudomonas putida* KT2440 and *Acinetobacter baylyi* ADP1) (40, 47). In PS-11, such redundancy may ensure robust expression of 4-HBA hydroxylase under diverse environmental conditions, potentially broadening substrate utilization. Downstream of these *pob* clusters, a complete protocatechuate operon was found, encoding enzymes of the protocatechuic acid 4,5-cleavage pathway, including protocatechuic acid 4,5-dioxygenase, 2-pyrone-4,6-dicarboxylate hydrolase, 4-carboxy-2-hydroxymuconate-6-semialdehyde dehydrogenase, and associated hydratase and aldolase enzymes (Fig. 5F). This indicates that PS-11 employs the 4,5-cleavage route for protocatechuate, previously reported in *Comamonas testosteroni* and *Sphingobium* spp. (48, 49), but not yet described in detail for *Delftia*. Pathway modules with reference to xenobiotics and aromatics compound degradation found in the genome of *Delftia* strain PS-11 are presented in Table S2 (https://figshare.com/s/479569d8fb80f976b8ea).

Collectively, these observations suggest that PS-11 possesses a dual hydroxybenzoic acid funneling strategy, channeling salicylate and 3-HBA into the gentisic acid pathway,

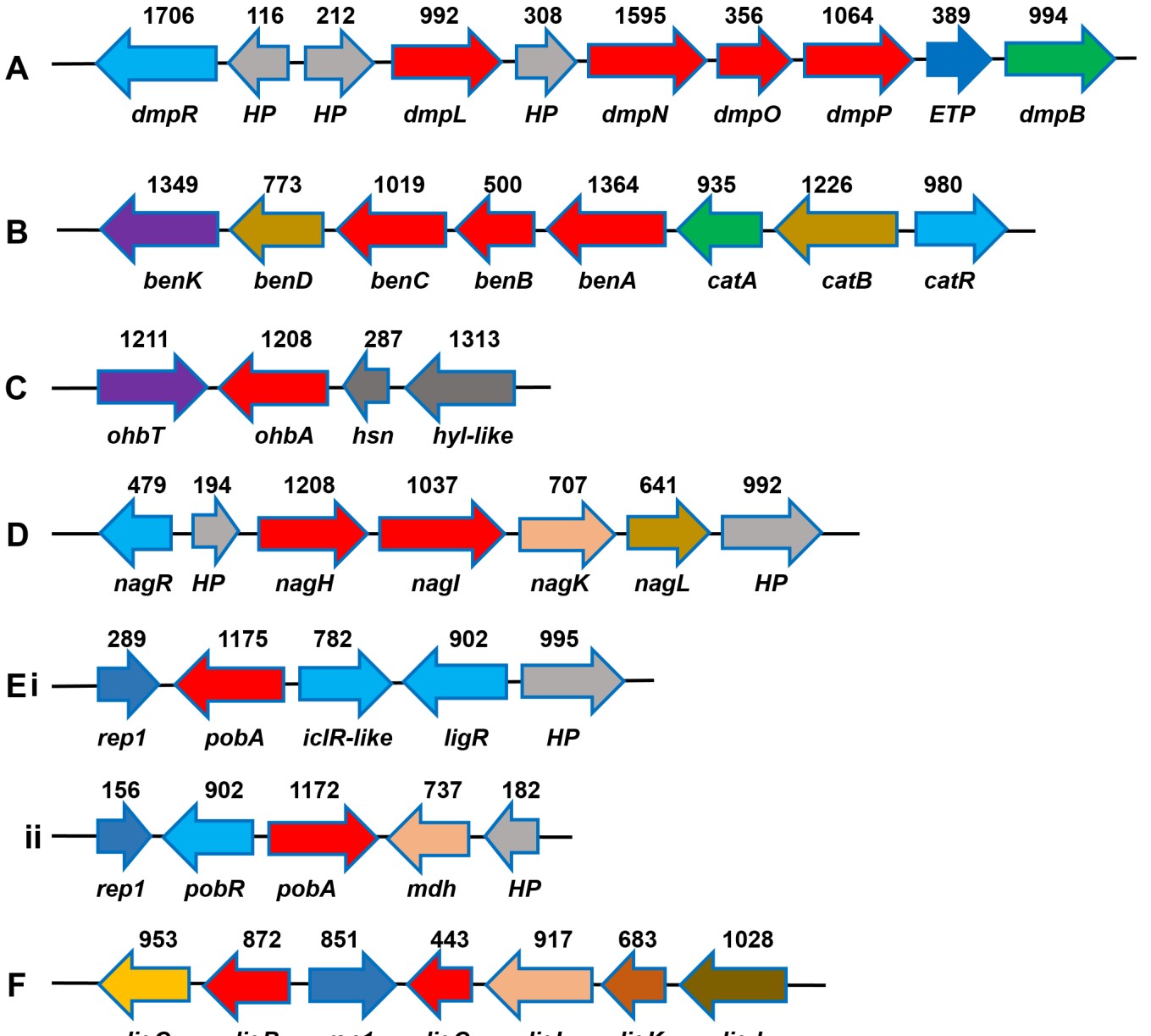

**FIG 5** Gene organization of aromatic degradation pathways from *Delftia* strain PS-11. Gene length (bp) and names are indicated above and below the gene symbols, respectively, which are depicted as filled single-headed arrows. Arrow directions indicate the orientation of genes on the chromosome. (A) Phenol pathway: *dmpR*, regulator of phenol hydroxylase; *HP*, hypothetical protein; *dmpL*, phenol hydroxylase, P1 oxygenase component; *dmpN*, phenol hydroxylase, P3 oxygenase component; *dmpO*, phenol hydroxylase, P4 oxygenase component; *dmpP*, phenol hydroxylase, FAD- and [2Fe-2S]-containing reductase component; *ETP*, iron-sulfur binding electron transfer protein; *dmpB*, catechol 2,3-dioxygenase. (B) Benzoic acid pathway: *benK*, benzoate MFS transporter; *benD*, 1,2-dihydroxycyclohexa-3,5-diene-1-carboxylate dehydrogenase; *benC*, benzoate 1,2-dioxygenase, ferredoxin reductase component; *benB*, benzoate 1,2-dioxygenase beta subunit; *benA*, benzoate 1,2-dioxygenase alpha subunit; *catA*, catechol 1,2-dioxygenase; catB, muconate cycloisomerase; *catR*, aromatic hydrocarbon utilization transcriptional regulator (LysR family). (C) Salicylic acid pathway: *ohbT*, MFS-type transporter; *ohbA*, putative n-hydroxybenzoic acid hydroxylase; *hsn*, DNA-binding protein H-NS; *hyl-like*, hemolysins and related proteins containing CBS. (D) 3-Hydroxybenzoic acid pathway: *nagR*, transcriptional regulator (MarR family); *nagH*, 3-hydroxybenzoate 6-monooxygenase; *nagI*, gentisate 1,2-dioxygenase; *nagK*, 3-fumarylpyruvate hydrolase; *nagL*, maleylpyruvate isomerase. (E) 4-Hydroxybenzoate pathway: (i) *rep1*, repeat region; *pobA*, P-hydroxybenzoic acid hydroxylase; *iclR-like*, transcriptional regulator; *ligR*, transcriptional regulator (LysR family), (ii) *pobR*, transcriptional regulator (AraC family); *pobA*, P-hydroxybenzoic acid hydroxylase; *mdh*, putative metal-dependent hydrolase. (F) Protocatechuate pathway: *ligC*, 4-carboxy-2-hydroxymuconate-6-semialdehyde dehydrogenase; *ligB*, protocatechuate 4,5-dioxygenase beta chain; *ligC*, protocatechuate 4,5-dioxygenase alpha chain; *ligG*, 2-pyrone-4,6-dicarboxylic acid hydrolase; *ligK*, 4-carboxy-4-hydroxy-2-oxoadipate aldolase; *ligJ*, 4-oxalomesaconate hydratase.

while 4-HBA is degraded via protocatechuic acid through the 4,5-cleavage route (50, 51). Such metabolic versatility likely expands the ecological niche of PS-11, enabling degradation of a wide spectrum of lignin-derived and anthropogenic hydroxyaromatics.

## Conclusion

The growth and presence of the active enzymes suggest that *Delftia* strain PS-11 has the capability to degrade aromatic compounds such as phenol and benzoic acid via diverse metabolic routes. The *Delftia* PS-11 is a promising candidate for bioremediation due to its xenobiotic-degrading potential, making it suitable for applications like the cleanup of contaminated land and water, wastewater treatment, and waste management. Moreover, genetic manipulation of strain PS-11 also allows its application in biosensor development for rapid, cost-effective environmental monitoring of specific xenobiotics. Furthermore, the existence of a unique gene pool specific to the *Delftia* strain PS-11 may confer its ability to degrade aromatic compounds present in the aquatic ecosystem and detoxify the environment for sustainability of underwater fauna and flora. Hence, *Delftia* strain from the freshwater pufferfish skin adds a new ecological niche to this bacterium.

## MATERIALS AND METHODS

### Bacterial culture, media, and growth conditions

The strain PS-11 was isolated from the surface skin mucus of pufferfish. The pufferfish samples were collected from the Mahanadi River, India (coordinates: 20°26946.60N 85°44928.30E), in August 2018 and were subsequently transported to the laboratory in plastic containers filled with river water. The mucus from the pufferfish skins was collected with sterile cotton swabs and put into 1 mL sterile phosphate-buffered saline (PBS, pH 7.4), mixed by vortexing and used as an original inoculum/sample (52). A 100 µL aliquot was serially diluted using PBS and plated onto nutrient agar (BD, Difco). All plates were incubated at 30°C corresponding to the river water temperature for 2 days. Several colonies that appeared on nutrient agar were picked and purified. A small cream-colored colony, designated strain PS-11, was selected for further analysis. Cultures were maintained on nutrient agar (BD, Difco) and stored at 4°C for short-term preservation. Cultures were stored at −80°C in 15% (vol/v) glycerol for long-term preservation.

*Delftia* strain PS-11 was grown in 150 mL M9 minimal medium (HiMedia, India) supplemented aseptically (wt/vol) with benzene (0.05%), phenol (0.05%), benzoic acid (0.05%), or 2-/3-/4-hydroxybenzoic acid (0.05%) in 500 mL baffled Erlenmeyer flasks at 30°C on a rotary shaker (200 rpm). Growth was monitored every 2 h by measuring optical density at 540 nm using a spectrophotometer (JASCO V-530, Japan).

### Phenotypic characterization

Gram staining was carried out using the commercial kit (HIMEDIA, India). Oxidase activity was assessed with discs impregnated with dimethyl *p*-phenylenediamine (HIMEDIA, India). Catalase activity was assayed by mixing a pellet of a freshly centrifuged culture with a drop of hydrogen peroxide (10%, vol/vol). Anaerobic growth was determined on Nutrient agar (BD, Difco) and on nutrient agar supplemented with potassium nitrate (0.1%, wt/vol) in an anaerobic jar using the BD GasPak EZ system (Becton Dickinson, USA). The jar was incubated at 37°C and examined after 7 days. Utilization of different sugars was tested using HiCarbo Kit (HIMEDIA, India).

### Identification of bacteria by 16S rRNA gene sequencing

Genomic DNA was extracted following the methods of Sambrook and Russell (53), and PCR was carried out using the universal bacterial primers 27F (5′-GAGTTTGATCCTGGCT CAG-3′) and 1525R (5′-AAAGGAGGTGATCCAGCC-3′) (54). The PCR product was purified using a QIAQuick Gel Extraction Kit (Qiagen, Germany) and sequenced in a capillary DNA

analyzer (3500, Applied Biosystems) following the manufacturer's protocol. The 16S rRNA gene sequences were assembled using the sequence alignment editor program BioEdit (55) and compared with closely related type strains retrieved from GenBank after BLAST searches (56) and using the EzBioCloud Database (57).

## Whole-genome sequencing and annotation

The genomic DNA of *Delftia* strain PS-11 was isolated using standard methods (53). The DNA concentration and quality were measured using a NanoDrop 8000 spectrophotometer (Thermo Fisher Scientific). A combination of short-read Illumina and long-read Oxford Nanopore Technologies sequencing platforms was used to generate the complete genome sequence of *Delftia* strain PS-11. For high-throughput sequencing, a TG TruSeq Nano DNA HT library Preparation Kit (Illumina) was used for constructing the paired-end sequencing library from 200 ng of DNA. The quality and quantity of the library were checked by using Bioanalyzer 2100 (Agilent Technologies) as per the manufacturer's instructions. Illumina short-read DNA sequencing was carried out using the Illumina HiSeq 4000 next-generation sequencing platform to produce paired-end sequence reads with 150-bp read length. Read quality was assessed with the FastQC v0.11.5 program (http://www.bioinformatics.babraham.ac.uk/projects/fastqc) using default parameters. Low-quality bases and adapters were removed with the Trimmomatic v0.36 (58) using the parameters mentioned in the manual. Trimmed reads were assembled *de novo* using the Unicycler v0.4.9 (59), followed by assembly polishing with the Pilon v1.23 (60). Assembly quality was assessed with the QUAST v4.6.1 (61). For long-read Nanopore sequencing, a genomic library was prepared using the Nanopore ligation sequencing kit (SQK-LSK109; Oxford Nanopore Technologies, Oxford, UK). The library was sequenced with an R9.4.1 MinION flow cell (FLO-MIN106) using the MinKNOW v2.0 with default settings. Barcode and adapter sequences were trimmed from Nanopore long reads using Porechop v0.2 (https://github.com/rrwick/Porechop), and reads with a minimum length of 1 kb were filtered using seqtk v1.2 (https://github.com/lh3/seqtk) for downstream analysis. The hybrid genome assembly was performed using the Unicycler v0.4.9 in hybrid assembly mode (59). The highly accurate Illumina short reads were aligned against the long Nanopore reads to sort out random sequencing errors (59). The assembled genomes were annotated using the NCBI Prokaryotic Genome Annotation Pipeline v4.9 with default parameters (62). The completeness and contamination of the whole-genome sequence were measured using the CheckM (63). The genomic GC content and assembly statistics were determined using a Perl script (https://github.com/tomdeman-bio/Sequence-scripts/blob/master/calc_N50_GC_Genomesize.pl). In addition, the RAST Server operates the GLIMMER algorithm, which was used to predict the protein-coding genes (64, 65).

## Comparative genomics and phylogenetic analysis

The genomic relatedness of strain PS-11 was compared with the reference genomes of four closely related *Delftia* type species that are available in the National Center for Biotechnology Information (NCBI) database (last accessed 20 March 2025). The whole-genome sequence of reference strains compared in this study is listed in (Table 1). The ANI was calculated using the python module pyani (https://github.com/widdowquinn/pyani) method. The isDDH similarity was measured with the help of the Genome-to-Genome distance calculator (22). The AAI was estimated using the "aai_wf" function implemented in the compareM program (https://github.com/dparks1134/CompareM). The 16S rRNA gene sequence-based phylogenetic tree was constructed according to the Kimura two-parameter model (66) using the MEGA version 11 (67) software package (Biodesign Institute, Arizona, USA). For phylogenomic analysis using the genome-wide core genes, type species of *Delftia* with more than 95% genome completeness along with 16S rRNA gene sequence similarity values more than 95% were retrieved from the NCBI database (https://github.com/kblin/ncbi-genome-download/) and compared in this study. The core genes extracted by the UBCG pipeline (68) were concatenated, and a

maximum-likelihood tree was constructed with the GTR model using the RAxML tool (69).

Additionally, a CGView comparison tool was used to graphically represent the BLAST comparison of the three *Delftia* species genomes against strain *Delftia* PS-11 genome (70). Clusters of Orthologous Genes (COGs) have been identified using Reverse Position-Specific BLAST against NCBI's Conserved Domain Database with an *e*-value cut-off of 0.00,001 (71). COG functional category information was obtained using cdd2cog version 0.1 (71). To determine the KEGG class, we leveraged the "anvi-run-kegg-kofams" script embedded within the Anvio program to perform an HMM search against the KEGG orthologs database (72). The KEGG functional annotations for all four *Delftia* genomes were generated using the same "anvi-run-kegg-kofams" script to ensure consistency in functional assignment. The distribution of COGs and KEGG functional enrichment across *Delftia* genomes was visualized in a heatmap generated using the ggplot2 and pheatmap R packages, implemented in R version 4.0.4 (73). The pangenome analysis and identification of species-specific genes for four *Delftia* isolates were conducted using the Anvi'o pipeline (version 7.1) (74). Gene clusters were identified through the Markov cluster algorithm (MCL) algorithm with an inflation parameter of 10, optimized for sensitivity in closely related genomes (75). Single-copy core genes were identified from gene clusters using the "anvi-get-sequences-for-gene-clusters" script within the Anvi'o software suite (74).

## Preparation of cell-free extracts and enzyme assays

### Preparation of cell-free extract

The cell-free extract (CFE) was prepared from cells grown on various aromatic compounds. Cells were harvested by centrifugation and washed twice with ice-cold potassium phosphate buffer (50 mM, pH 7.5) or Tris-HCl buffer (50 mM, pH 7.5) and re-suspended in the same buffer (1:10, wt/vol). Cells were lysed on ice by using an ultrasonic processor (GE130, USA) (4 cycles with 15 pulse each [1 s pulse, 1 s interval, cycle duration 30 s, output 15W] with a 5 min intervals between each cycle). The cell homogenate was centrifuged at 30,000 × *g* for 30 min. The clear supernatant obtained was referred to as cell-free extract (CFE). Protein concentration was estimated as described by Bradford 1976 (76) using bovine serum albumin as the standard.

### Enzyme assays

All enzyme activities were monitored spectrophotometrically (Lambda 35, Perkin Elmer, USA). To decipher the mode of ring-cleavage by protocatechuate dioxygenase, time-dependent spectral changes (in the range of 220 nm to 500 nm) were monitored in phosphate/Tris-Cl buffer (50 mM, pH 7.5) with 3,4-dihydroxybenzoic acid (3,4-DHB; 100 µM) and appropriate amounts of CFE (77). Protocatechuate 3,4-dioxygenase (P34DO) activity was monitored by measuring the rate of decrease in absorbance at 290 nm due to disappearance of substrate 3,4-dihydroxybenzoic acid (3,4-DHB, $\varepsilon_{290} = 3{,}870$ M$^{-1}$ cm$^{-1}$) (77). The reaction mixture (1 mL) contained 3,4-DHB (100 µM), appropriate amount of CFE, and phosphate/Tris-Cl buffer (50 mM, pH 7.5). Protocatechuate 4,5-dioxygenase (P45DO) activity was monitored by measuring the rate of increase in absorbance at 410 nm due to formation of product α-hydroxy-γ-carboxymuconic semialdehyde ($\varepsilon_{410} = 17{,}200$ M$^{-1}$ cm$^{-1}$ (78). The reaction mixture (1 mL) contained 3,4-DHB (100 µM), appropriate amount of CFE, and phosphate/Tris-Cl buffer (50 mM, pH 7.5).

Catechol 1,2-dioxygenase (C12DO) activity was monitored by measuring the rate of formation of *cis,cis*-muconic acid at 260 nm ($\varepsilon_{260} = 16{,}000$ M$^{-1}$ cm$^{-1}$ (79). The reaction mixture (1 mL) contained catechol (100 µM), appropriate amount of CFE, and phosphate/Tris-Cl buffer (50 mM, pH 7.5). Catechol 2,3-dioxygenase (C23DO) activity was monitored by measuring the rate of formation of yellow product *cis*-muconic semialdehyde at 375 nm ($\varepsilon_{375} = 33{,}000$ M$^{-1}$ cm$^{-1}$ (80). The reaction mixture (1 mL) contained

catechol (100 µM), appropriate amount of CFE, and phosphate/Tris-Cl buffer (50 mM, pH 7.5).

Gentisate 1,2-dioxygenase (G12DO) activity was monitored by measuring the rate of formation of maleylpyruvic acid at 330 nm ($\varepsilon_{330}$ = 13,000 $M^{-1}$ $cm^{-1}$ (81). The reaction mixture (1 mL) contained gentisic acid (100 µM), appropriate amount of CFE, and phosphate/Tris-Cl buffer (50 mM, pH 7.5). The specific activity of enzymes was expressed as nmol $min^{-1}$ $mg^{-1}$ protein in the CFE. All enzymatic assays were performed using three independent biological replicates, each measured in triplicate, and results are presented as mean ± standard deviation.

## ACKNOWLEDGMENTS

R.R.A.K., S.D., and T.D. acknowledge the Council of Scientific and Industrial Research and Department of Biotechnology, New Delhi, respectively, for providing the research fellowship. The authors acknowledge the Distributed Information Sub-Center at the Institute of Life Sciences, Bhubaneswar, for the computational facility.

This work was supported in part by the funding received from the Ministry of Earth Sciences, Government of India (F. No. MoES/PAMC/DOM/181/2023 (E-14616) to Dr. Debasis Dash, Director, Institute of Life Sciences, Bhubaneswar. This work was carried out by using the institutional research facility. This research received no specific grant from any funding agency in the public, commercial, or not-for-profit sectors.

S.K.D. developed the concept. S.K.D. and P.S.P. coordinated the experiments and designed the experiments. R.R.A.K., S.D., T.D., and S.P. participated in experiments. R.R.A.K., S.D., S.P., P.S.P., and S.K.D. interpreted the data and wrote the manuscript. Dr. Debasis Dash, Director, ILS, Bhubaneswar, critically read the manuscript and provided the infrastructure facility. All authors read and approved the final manuscript.

## AUTHOR AFFILIATIONS

[1]Department of Biotechnology, BRIC-Institute of Life Sciences, Bhubaneswar, India
[2]Department of Biosciences and Bioengineering, Indian Institute of Technology Bombay, Mumbai, Maharashtra, India

## PRESENT ADDRESS

Sushanta Deb, Department of Veterinary Microbiology and Pathology, College of Veterinary Medicine, Washington State University, Pullman, Washington, USA
Tanmoy Debnath, Immunology Laboratory, CSIR-Institute of Microbial Technology, Chandigarh, India

## AUTHOR ORCIDs

Prashant S. Phale http://orcid.org/0000-0002-5781-8941
Subrata K. Das http://orcid.org/0000-0002-7280-5751

## AUTHOR CONTRIBUTIONS

Ritu Rani Archana Kujur, Formal analysis, Investigation, Methodology, Writing – original draft | Sushanta Deb, Data curation, Formal analysis, Investigation, Methodology, Software, Validation, Writing – original draft, Writing – review and editing | Tanmoy Debnath, Formal analysis, Investigation, Methodology, Validation | Sandesh Papade, Data curation, Formal analysis, Investigation, Methodology, Validation, Writing – original draft, Writing – review and editing | Prashant S. Phale, Conceptualization, Data curation, Formal analysis, Investigation, Methodology, Supervision, Validation, Writing – original draft, Writing – review and editing | Subrata K. Das, Conceptualization, Data curation, Formal analysis, Investigation, Methodology, Project administration, Resources, Software,

Supervision, Validation, Visualization, Writing – original draft, Writing – review and editing

## DATA AVAILABILITY

The GenBank/EMBL/DDBJ accession number for the genome and 16S rRNA gene sequences of Delftia strain PS-11 are JACSYA000000000 and MK165140, respectively.

## ETHICS APPROVAL

This study was carried out with approval from the Institutional Animal Ethics Committee (Letter No. V-311-MISC/2017-18/ILS/884).

## ADDITIONAL FILES

The following material is available online.

Open Peer Review

**PEER REVIEW HISTORY (review-history.pdf).** An accounting of the reviewer comments and feedback.

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
