## [Reviewer comments · Microbiology Spectrum]

Microbiology Spectrum

Genomic and functional insights into aromatic compound degradation by *Delftia* strain PS-11 isolated from freshwater pufferfish

Ritu Archana Kujur, Sushanta Deb, Tanmoy Debnath, Sandesh Papade, Prashant Phale, and Subrata Das

Corresponding Author(s): Subrata Das, Institute of Life Sciences

Review Timeline:

Submission Date:	September 28, 2025
Editorial Decision:	December 16, 2025
Revision Received:	January 16, 2026
Editorial Decision:	February 6, 2026
Revision Received:	February 27, 2026
Accepted:	March 2, 2026

Editor: Dirk Tischler

Reviewer(s): Disclosure of reviewer identity is with reference to reviewer comments included in decision letter(s). The following individuals involved in review of your submission have agreed to reveal their identity: Adebayo Jonathan Adeyemo (Reviewer #1); Yvonne Marvellous Akpudo (Reviewer #2); Vida Časaitė (Reviewer #3)

Transaction Report:

DOI: <https://doi.org/10.1128/spectrum.03105-25>

Re: Spectrum03105-25 (Genomic and functional insights into aromatic compound degradation by *Delftia* strain PS-11 isolated from freshwater pufferfish)

Dear Prof. Subrata Das:

Thank you for the privilege of reviewing your work. Below you will find my comments, instructions from the Spectrum editorial office, and the reviewer comments.

Revision Guidelines

Sincerely,
Dirk Tischler
Editor
Microbiology Spectrum

Reviewer #1 (Comments for the Author):

The manuscript presents valuable genomic and functional insights into *Delftia* strain PS-11 and its aromatic compound degradation pathways. The study is novel, well-structured, and scientifically relevant. However, several areas require minor but important improvements:

Include quantitative results (e.g., enzyme activity or degradation rate) to strengthen claims in the abstract.

Condense background information and include recent literature should be introduction between 2022-2025 on *Delftia* and biodegradation.

Provide genome assembly statistics in the result in discussion, clearly report units and replicates for enzymatic assays, and consider adding a schematic pathway figure.

In the statistical analysis, specify the exact test used for mean separation and ensure consistency across tables and text.

Conclusion should highlight the biotechnological or environmental application potential of *Delftia* PS-11.

in the Methods & References, Maintain consistency in software citation, include replication details, and update outdated references.

Overall, this is a strong paper requiring minor revision for clarity, methodological precision, and updated literature support before acceptance.

Reviewer #2 (Public repository details (Required)):

Datasets have already been deposited in public repositories including supplementary materials in figshare, and genomic data in NCBI database.

Reviewer #2 (Comments for the Author):

Recent studies have demonstrated active research on *Delftia* sp underscoring ecological versatility based on diverse metabolic capabilities consistent with an increasingly large pan-genome sizes as more genes are discovered. While this manuscript expands on similar context, it lacks the acknowledgement of previously identified *Delftia* sp from closely related niche / Fish, e.g. from skin mucus of Zebrafish (refer to Genome announcement: Draft Genome Sequence of *Delftia* tsuruhatsensis Strain 45.2.2, Colonizer of Zebrafish, *Danio rerio*, Skin Mucus - PMC), and other organs of Nile tilapia (<https://doi.org/10.3389/fmicb.2023.1321122>) (Antimicrobial Resistance analysis of Pathogenic Bacteria Isolated from Freshwater Nile Tilapia (*Oreochromis niloticus*) Cultured in Kerala, India | Current Microbiology). Take these into account since only references on *Delftia* sp from soil, constructed wetland, and sea were cited; and one of the notable conclusion of this paper is that "*Delftia* strain from the freshwater pufferfish skin adds a new ecological niche to this bacterium". I would suggest further search and confirmation with existing literature that highlight *Delftia* sp as one of the colonizers of any organs of other freshwater fish sp.

Line 55 - "Therefore, this study adds a new ecological versatility of this bacterium." How accurate is this statement? Clarify the meaning of 'new ecological versatility' implied. Bear in mind that previous research have identified *Delftia* sp as a generalist with the capability to utilize diverse sources of carbon including environmental organic pollutants such as phenolic compounds / benzoate derivatives.

In the methodology, it may be expedient to precisely include the name of the river from which the host (pufferfish) of the isolated bacteria was sampled regardless of cited reference. More so, it may be informative to further clarify if the described strain was isolated from the same host of the other isolate (*Vibrio cholerae* - ref 56) or another pufferfish sampled from the same river. This may be helpful to readers if they try to link both studies in relation to potential contamination of freshwater body since none of these papers hinted at such while demonstrating evidence of distinct organisms with delineated implications linked to ecological health and integrity.

Additionally, consider rephrasing few statements in the lines below;

Line 85- "...plenty of studies of the metabolic pathways..."

Line 164 - " The similar KEGG based approach..."

Reviewer's Comments for the Manuscript titles " Genomic and functional insights into aromatic compound degradation by Delftia strain PS-11 isolated from freshwater pufferfish

In the **Abstract, between the (Lines 39–63)**

The abstract is structured very well, but overly descriptive is observed. It would benefit from the following quantitative highlights (e.g., specific enzyme activity values or degradation rates).

Rephrase lines 41–42 for improved readability: "The genome of strain PS-11 was sequenced in order to understand..." can be substituted with "Whole-genome sequencing revealed the catabolic processes of xenobiotic degradation."

Lines 53–54: The novelty claim ("presence of Delftia strain from pufferfish is not yet reported") is strong; a reference in support might be added.

Introduction (Lines 74–115): Introduction is good in providing general background but is too long. Consider shortening generic study references (Lines 90–99).

Lines 100–107: Very good justification of Delftia relevance; however, add recent references (2023–2025) on Delftia biodegradation.

Lines 110–115: Objectives are clearly formulated but can be stronger if presented as direct research questions.

Between Lines 117–328 in **Results and Discussion**. Lines 118–126, Phenotypic description is adequate but leave out comparative morphology with well-related Delftia strains.

In lines 128–133, include assembly metrics (coverage, N50, contig count).

In lines 135–150, species delineation discussion is accurate but should utilize average nucleotide identity thresholds (95–96 %) with current taxonomic recommendations (e.g., GTDB 2023).

In lines 152–173, the comparative genomics is thorough but would be helpfully summarized as a table contrasting Delftia PS-11 with other strains by xenobiotic gene clusters.

In lines 190–214, enzyme assay interpretation is well done but raw specific activities (nmol/min/mg) should be presented in the main text or supplementary data.

In lines 217–320, the pathway descriptions (phenol, benzoate, hydroxybenzoate) are rich in detail; however, the description could be improved by including schematic figures linking genes to metabolites for readability.

In lines 243–258, elaborate gene nomenclature consistency (italicize all gene names, e.g., *dmpR*, *catA*, *pobA*).

In lines 275–283, "A remarkable aspect of the PS-11 operon is the presence of RpoH...", this subsection should include supporting literature to enhance the novelty claim.

In lines 317–320, there is excellent integration of double hydroxybenzoate degradation; however, do mention at least one recent comparative Delftia genomic study (2024–2025).

In the **conclusion section** in lines 322–328, the conclusion adequately sums up results but can be stronger by the highlighting of bioremediation potential and industrial/environmental relevance.

Materials and Methods between the Lines 330–466, Complete and reproducible.

In lines 366–392, use standard software versions consistently (e.g., Unicycler v0.4.9 vs v0.4.7 used previously).

Also in lines 432–462, include replication number and statistical tests used to verify enzyme activity (e.g., standard deviation or ANOVA).

Acknowledgements & Author Contributions (Lines 468–484):

Clear and transparent. No ethical or conflict concerns identified.

Line 475: The statement “This research received no specific grant...” contradicts earlier funding acknowledgements; clarify funding scope.

References (Lines 515–757):

Comprehensive, but older citations (2000–2010) dominate the list. Incorporate more recent literature (2022–2025) on microbial aromatic degradation and Delftia genomics.

Verify DOI formatting (e.g., Lines 631–637 contain line breaks within DOIs).

Overall Evaluation The manuscript presents a robust genomic and enzymatic characterization of Delftia PS-11, an ecologically novel isolate. However, the paper requires minor structural and consistency revisions, particularly in data presentation (quantitative details, table clarity, gene name formatting, and literature updates).

Recommendation: Minor Revision (with focus on clarity, consistency, and recent citations).

Acknowledgements & Author Contributions

Clear and transparent. No ethical or conflict issue detected.

Line 475: The statement "This research received no specific grant..." contradicts earlier funding acknowledgements; please clarify the funding extent.

References

Comprehensive, but older citations (2000–2010) dominate the list. Include recent publications (2022–2025) on microbial aromatic degradation and Delftia genomics.

Verify DOI formatting (e.g., Lines 631–637 contain line breaks within DOIs).

Overall Evaluation The manuscript is a solid genomic and enzymatic characterization of *Delftia* PS-11, an ecologically novel isolate. The manuscript does require some minor structural and consistency revisions, though, particularly in data presentation (quantitative details, table legibility, gene nomenclature format, and literature timeliness).

Response to the reviewer's comments

We are grateful to the editor and reviewers for reviewing our manuscript and providing us necessary comments and suggestions for improving our manuscript. Please find below Response to the reviewer's comments for your kind consideration

Reviewer #1 (Comments for the Author):

The manuscript presents valuable genomic and functional insights into *Delftia* strain PS-11 and its aromatic compound degradation pathways. The study is novel, well-structured, and scientifically relevant. However, several areas require minor but important improvements:

Reply: We appreciate the reviewer's comments. As suggested, we have revised the manuscript accordingly.

Include quantitative results (e.g., enzyme activity or degradation rate) to strengthen claims in the abstract.

Reply: Following the reviewer's guidance, we have incorporated specific enzyme activity values in the abstract of the revised manuscript.

Condense background information and include recent literature should be introduction between 2022-2025 on *Delftia* and biodegradation.

Reply: We appreciate the reviewer's comments. As suggested, we have revised the manuscript accordingly.

Provide genome assembly statistics in the result in discussion, clearly report units and replicates for enzymatic assays, and consider adding a schematic pathway figure.

Reply: We appreciate the reviewer's comments. A total of 47,061,338 base pair reads were obtained through whole-genome sequencing. The lengths of the forward and reverse reads were 23,450,336 bp each. Sequence alignment against the reference genome resulted in 99.75% of sequences mapped, of which 98.09% were correctly oriented and placed as paired ends. The hybrid genome assembly resulted in a continuous, non-fragmented single contig spanning 5,538,489 bp, achieving an N50 value equal to that length and covering 81.32% of the genome. It has been incorporated in the revised manuscript. Because the *Delftia* PS-11 genome was assembled into a single, continuous unit through hybrid assembly, the final genome assembly is free of the variable contig lengths that typically generate contig statistics.

We have now clearly specified the units of enzyme activity and the number of biological and technical replicates for all enzymatic assays in the Materials and Methods section, as well as in the relevant text and tables. Enzyme activities are reported as $\text{nmol min}^{-1} \text{mg}^{-1}$ protein. All assays were conducted using three independent biological replicates, each measured in triplicate (technical replicates), and results are presented as mean \pm standard deviation (SD). In addition, we have revised Figure 4 to include a schematic representation of the proposed enzymatic degradation pathways of aromatics in *Delftia* sp. PS-11,

highlighting key metabolites, genes, enzymes, and experimentally validated enzyme activities to improve readability and facilitate understanding.

In the statistical analysis, specify the exact test used for mean separation and ensure consistency across tables and text.

Reply: We appreciate the reviewer's concern. However, in the present study, mean separation (i.e., post-hoc multiple comparison of means) was not performed, as it was not required for the analysis or interpretation of the data presented. The statistical evaluation was therefore limited to descriptive statistics / overall comparison, which was sufficient to support the conclusions drawn.

Conclusion should highlight the biotechnological or environmental application potential of *Delftia* PS-11.

Reply: We appreciate the reviewer's comments. As suggested, we have incorporated the biotechnological or environmental application potential of *Delftia* PS-11 in the conclusion part of the revised manuscript.

in the Methods & References, Maintain consistency in software citation, include replication details, and update outdated references.

Reply: Following the reviewer's guidance, necessary rectification have been made in the revised manuscript.

Overall, this is a strong paper requiring minor revision for clarity, methodological precision, and updated literature support before acceptance.

Reply: We appreciate the reviewer's comments. As suggested, we have revised the manuscript accordingly.

Reviewer's Comments for the Manuscript titles "Genomic and functional insights into aromatic compound degradation by *Delftia* strain PS-11 isolated from freshwater pufferfish

In the Abstract, between the (Lines 39–63)

The abstract is structured very well, but overly descriptive is observed. It would benefit from the following quantitative highlights (e.g., specific enzyme activity values or degradation rates).

Reply: We appreciate the reviewer's comments. As suggested, we have incorporated specific enzyme activity values in the abstract of the revised manuscript.

Rephrase lines 41–42 for improved readability: "The genome of strain PS-11 was sequenced in order to understand..." can be substituted with "Whole-genome sequencing revealed the catabolic processes of xenobiotic degradation."

Reply: We appreciate the reviewer's comments. As suggested, lines 41–42 has been rephrased in the revised manuscript.

Lines 53–54: The novelty claim ("presence of *Delftia* strain from pufferfish is not yet reported") is strong; a reference in support might be added.

Reply: Earlier studies have demonstrated the presence of *Delftia* species in the European eel, *Anguilla anguilla* (L.) (**Andree et al. 2013**. Co-infection with *Pseudomonas anguilliseptica* and *Delftia acetivorans* in the European eel, *Anguilla anguilla* (L.): a case history of an illegally trafficked protected species. *J Fish Dis.* 36(7):647-56. doi: 10.1111/jfd.12066). Also reported from the injured eye of freshwater Nile Tilapia (*Oreochromis niloticus*) (**Preena et al. 2020**. Antimicrobial Resistance analysis of Pathogenic Bacteria Isolated from Freshwater Nile Tilapia (*Oreochromis niloticus*) Cultured in Kerala, India. *Curr Microbiol.* 77(11):3278-3287. doi: 10.1007/s00284-020-02158-1). However, the presence of *Delftia* strain from pufferfish is not yet reported that justify the novelty claim.

Introduction (Lines 74–115): Introduction is good in providing general background but is too long. Consider shortening generic study references (Lines 90–99).

Reply: We appreciate the reviewer's comments. As suggested, lines 90-99 has been rephrased in the revised manuscript.

Lines 100–107: Very good justification of *Delftia* relevance; however, add recent references (2023–2025) on *Delftia* biodegradation.

Reply: We appreciate the reviewer's comments. As suggested, the following recent references have been incorporated in the revised manuscript.

Chen R, Zhao Z, Xu T, Jia X. **2023**. Microbial Consortium HJ-SH with Very High Degradation Efficiency of Phenanthrene. *Microorganisms.* 11(10):2383. doi: 10.3390/microorganisms11102383.

Li S, Geng Y, Bao C, Mei Q, Shi T, Ma X, Hua R, Fang L. **2024**. Complete biodegradation of fungicide carboxin and its metabolite aniline by *Delftia* sp. HFL-1. *Sci Total Environ.* 912:168957. doi: 10.1016/j.scitotenv.2023.168957.

Farajollahi S, Lombardo NV, Crenshaw MD, Guo HB, Doherty ME, Davison TR, Steel JJ, Almand EA, Varaljay VA, Swei-Hung C, Mirau PA, Berry RJ, Kelley-Loughnane N, Dennis PB. **2024**. Defluorination of Organofluorine Compounds Using Dehalogenase Enzymes from *Delftia acidovorans* (D4B). *ACS Omega.* 9(26):28546-28555. doi: 10.1021/acsomega.4c02517.

Lines 110–115: Objectives are clearly formulated but can be stronger if presented as direct research questions.

Reply: We appreciate the reviewer's comments. The research objectives have been revised as suggested.

Between Lines 117–328 in Results and Discussion. Lines 118–126, Phenotypic description is adequate but leave out comparative morphology with well-related *Delftia* strains.

Reply: We appreciate the reviewer's comments. As suggested, we have deleted the comparative morphology with well-related *Delftia* strains from the revised manuscript.

In lines 128–133, include assembly metrics (coverage, N50, contig count).

Reply: We are grateful for the insightful comments. A total of 47,061,338 base pair reads were obtained through whole-genome sequencing. The lengths of the forward and reverse reads were 23,450,336 bp each. Sequence alignment against the reference genome resulted in 99.75% of sequences mapped, of which 98.09% were correctly oriented and placed as paired ends. The hybrid genome assembly resulted in a continuous, non-fragmented single contig spanning 5,538,489 bp, achieving an N50 value equal to that length and covering 81.32% of the genome. It has been incorporated in the revised manuscript.

In lines 135–150, species delineation discussion is accurate but should utilize average nucleotide identity thresholds (95–96 %) with current taxonomic recommendations (e.g. GTDB 2023).

Reply: Current taxonomic practice commonly applies an average nucleotide identity (ANI) threshold of approximately 95–96% as an operational species boundary; however, recent work emphasizes that this value does not represent a universal genetic discontinuity across prokaryotes but instead functions as a pragmatic guideline (**Parks et al. 2022**, *Nucleic Acids Research*, Volume 50, Issue D1, 7 January 2022, Pages D785–D794, <https://doi.org/10.1093/nar/gkab776>). An ANI threshold of 95–96% therefore remains the default and practical criterion for prokaryotic species delineation and is broadly consistent with current taxonomic recommendations). As suggested, we have used current taxonomic recommendations i.e GTDB and the average nucleotide identity threshold value remained the same.

In lines 152–173, the comparative genomics is thorough but would be helpfully summarized as a table contrasting *Delftia* PS-11 with other strains by xenobiotic gene clusters.

Reply: We are greatly appreciative of the valuable comments. In **Table 2**, comparative analysis of gene clusters involved in xenobiotics degradation of *Delftia* PS-11 with other strains based on KEGG pathway has been described in the revised manuscript. **Table 2**, has been re-numbered as **Table 3** in the revised manuscript.

In lines 190–214, enzyme assay interpretation is well done but raw specific activities (nmol/min/mg) should be presented in the main text or supplementary data.

Reply: We are grateful for the insightful comments, and have added the specific activities of enzymes (nmol/min/mg) to the revised manuscript as suggested.

In lines 217–320, the pathway descriptions (phenol, benzoate, hydroxybenzoate) are rich in detail; however, the description could be improved by including schematic figures linking genes to metabolites for readability.

Reply: We appreciate the reviewer's comment. We have revised Figure 4 to include a schematic representation of the proposed enzymatic degradation pathways of phenol, benzoate, and hydroxybenzoates in *Delftia* sp. PS-11. The updated figure links key metabolites with the corresponding genes and enzymes and highlights experimentally validated enzyme activities, thereby improving readability and facilitating better understanding for readers.

In lines 243–258, elaborate gene nomenclature consistency (italicize all gene names, e.g., *dmpR*, *catA*, *pobA*).

Reply: We appreciate the reviewer's comments. As suggested, gene names have been italicized in the revised manuscript.

In lines 275–283, "A remarkable aspect of the PS-11 operon is the presence of RpoH...", this subsection should include supporting literature to enhance the novelty claim.

Reply: We appreciate the reviewer's comments. It is established that the *rpoH* gene in *E. coli* produces the sigma 32 (σ^{32}) protein, a key factor that directs RNA polymerase to transcribe essential heat shock genes, effectively shielding cells from protein damage caused by sudden temperature rises or environmental stressors (**Grossman AD**, Zhou YN, Gross C, Heilig J, Christie GE, Calendar R. 1985. Mutations in the *rpoH* (*htpR*) gene of *Escherichia coli* K-12 phenotypically suppress a temperature-sensitive mutant defective in the sigma 70 subunit of RNA polymerase. *J Bacteriol.* 161(3):939-43. doi: 10.1128/jb.161.3.939-943.1985). The above explanation including supporting literature have been mentioned in the revised manuscript.

In lines 317–320, there is excellent integration of double hydroxybenzoate degradation; however, do mention at least one recent comparative *Delftia* genomic study (2024–2025).

Reply: We appreciate the reviewer's comments. As suggested the following two references have been included in the revised manuscript. (1), **Liu Y**, Zhao N, Dai S, He R, Zhang Y. 2024. Metagenomic insights into phenanthrene biodegradation in electrical field-governed biofilms for groundwater bioremediation. *J Hazard Mater.* 465:133477. doi: 10.1016/j.jhazmat.2024.133477. and (2), **Romero-Silva MJ**, Méndez V, Agulló L, Seeger M. 2013. Genomic and functional analyses of the gentisate and protocatechuate ring-cleavage pathways and related 3-hydroxybenzoate and 4-hydroxybenzoate peripheral pathways in *Burkholderia xenovorans* LB400. *PLoS One.* 8(2):e56038. doi: 10.1371/journal.pone.0056038.

In the conclusion section in lines 322–328, the conclusion adequately sums up results but can be stronger by the highlighting of bioremediation potential and industrial/environmental relevance.

Reply: We appreciate the reviewer's comments. As suggested, we have incorporated the biotechnological or environmental application potential of *Delftia* PS-11 in the conclusion part of the revised manuscript.

Materials and Methods between the Lines 330–466, Complete and reproducible.

Reply: We appreciate the reviewer's comments.

In lines 366–392, use standard software versions consistently (e.g., Unicycler v0.4.9 vs v0.4.7 used previously).

Reply: We are grateful to the reviewers. The standard software version Unicycler v0.4.9 is consistently referenced in the revised manuscript.

Also in lines 432–462, include replication number and statistical tests used to verify enzyme activity (e.g., standard deviation or ANOVA).

Reply: We appreciate the reviewer's concern. All enzyme activity assays were conducted using three independent biological replicates, each measured in triplicate (technical replicates), and the results are presented as mean \pm standard deviation (SD). In the present study, inferential statistical tests (e.g., ANOVA or t-test) were not applied, as mean separation or group-wise comparisons were not required for the analysis or interpretation of the data. Accordingly, the statistical evaluation was limited to descriptive statistics, which was sufficient to support the conclusions drawn.

Acknowledgements & Author Contributions (Lines 468–484):

Clear and transparent. No ethical or conflict concerns identified.

Reply: We appreciate the reviewer's comments

Line 475: The statement "This research received no specific grant..." contradicts earlier funding acknowledgements; clarify the funding extent.

Reply: Only monthly remuneration was paid to the author SKD from the fund received from the Ministry of Earth Sciences, Government of India (F. No. MoES/PAMC/DOM/181/2023 (E-14616). Other than remuneration, rest of the work was carried out using the basic research facility available in the Institute. Therefore, it is mentioned no specific research grant from any funding agency in the public, commercial, or not-for-profit sectors was received.

References (Lines 515–757):

Comprehensive, but older citations (2000–2010) dominate the list. Incorporate more recent literature (2022–2025) on microbial aromatic degradation and Delftia genomics.

Reply: We appreciate the reviewer's comments, Necessary changes with more recent literature have been incorporated in the revised manuscript.

Verify DOI formatting (e.g., Lines 631–637 contain line breaks within DOIs).

Reply: Necessary correction has been done in the revised manuscript.

Overall Evaluation The manuscript presents a robust genomic and enzymatic characterization of Delftia PS-11, an ecologically novel isolate. However, the paper requires minor structural and consistency revisions, particularly in data presentation (quantitative details, table clarity, gene name formatting, and literature updates).

Reply: We appreciate the reviewer's comments and encouragement. As suggested, necessary correction has been done in the manuscript.

Recommendation: Minor Revision (with focus on clarity, consistency, and recent citations).

Reply: We appreciate the reviewer's comments and encouragement. As suggested, necessary correction has been done in the revised manuscript.

Reviewer #2 (Public repository details (Required)):

Datasets have already been deposited in public repositories including supplementary materials in figshare, and genomic data in NCBI database.

Reply: We appreciate the reviewer's comments.

Reviewer #2 (Comments for the Author):

Recent studies have demonstrated active research on *Delftia* sp underscoring ecological versatility based on diverse metabolic capabilities consistent with an increasingly large pan-genome sizes as more genes are discovered. While this manuscript expands on similar context, it lacks the acknowledgement of previously identified *Delftia* sp from closely related niche / Fish, e.g. from skin mucus of Zebrafish (refer to Genome announcement: Draft Genome Sequence of *Delftia tsuruhatensis* Strain 45.2.2, Colonizer of Zebrafish, *Danio rerio*, Skin Mucus - PMC), and other organs of Nile tilapia (<https://doi.org/10.3389/fmicb.2023.1321122>) (Antimicrobial Resistance analysis of Pathogenic Bacteria Isolated from Freshwater Nile Tilapia (*Oreochromis niloticus*) Cultured in Kerala, India | Current Microbiology). Take these into account since only references on *Delftia* sp from soil, constructed wetland, and sea were cited; and one of the notable conclusion of this paper is that "*Delftia* strain from the freshwater pufferfish skin adds a new ecological niche to this bacterium". I would suggest further search and confirmation with existing literature that highlight *Delftia* sp as one of the colonizers of any organs of other freshwater fish sp.

Reply: We appreciate the reviewer's comments. As suggested relevant references have been acknowledged of previously identified *Delftia* sp from closely related niche / Fish in the revised manuscript.

Line 55 -"Therefore, this study adds a new ecological versatility of this bacterium." How accurate is this statement? Clarify the meaning of 'new ecological versatility' implied. Bear in mind that previous research have identified *Delftia* sp as a generalist with the capability to utilize diverse sources of carbon including environmental organic pollutants such as phenolic Compounds / benzoate derivatives.

Reply: We appreciate the reviewer's comments. We have revised and rephrased the statement accordingly.

In the methodology, it may be expedient to precisely include the name of the river from which the host (pufferfish) of the isolated bacteria was sampled regardless of cited reference. More so, it may be informative to further clarify if the described strain was isolated from the same host of the other isolate (*Vibrio cholerae* - ref 56) or another

pufferfish sampled from the same river. This may be helpful to readers if they try to link both studies in relation to potential contamination of freshwater body since none of these papers hinted at such while demonstrating evidence of distinct organisms with delineated implications linked to ecological health and integrity.

Reply: We appreciate the reviewer's comments. The pufferfish (*Tetraodon cutcutia*) samples were collected from Mahanadi River, India (coordinates: 20°26'46.60N 85°44'28.30E), in August 2018 and transported to the laboratory in a plastic container with river water. Mucus on pufferfish skin was taken using sterile cotton swabs and transferred into 1 mL of sterile phosphate-buffered saline (PBS), pH 7.4, to isolate bacteria. The bacteria from the cotton swabs were suspended in PBS by vigorous vortexing. The suspension was used as a master mix for the isolation of bacteria. An aliquot (100µl) of master mix sample was serially diluted using PBS (phosphate buffer saline) and plated onto nutrient agar (BD, Difco). All plates were incubated at 30°C corresponding to the river water temperature for 2 days. Several colonies developed at 30°C were picked and purified by repeated streaking on the same medium. Among them strain PS-11 was representing a genus *Delftia* based on 16S rRNA sequence analysis and was designated *Delftia* strain PS-11. Furthermore, the described strain was isolated at the same time from the same host of the other isolate (*Vibrio cholerae* - ref 56).

As suggested, it has been precisely included in the methodology section.

Additionally, consider rephrasing few statements in the lines below;
Line 85- "...plenty of studies of the metabolic pathways..."

Reply: We appreciate the reviewer's comments. We have rephrased the statements.

Line 164 - " The similar KEGG based approach..."

Reply: We appreciate the reviewer's comments. We have rephrased the statements.

Re: Spectrum03105-25R1 (Genomic and functional insights into aromatic compound degradation by *Delftia* strain PS-11 isolated from freshwater pufferfish)

Dear Prof. Subrata Das:

Thank you for the privilege of reviewing your work. Below you will find my comments, instructions from the Spectrum editorial office, and the reviewer comments.

Reviewer 3 found some minor issues which need to be addressed prior moving to the next step. Please make changes as suggested and I will make the final decision.

Revision Guidelines

Sincerely,
Dirk Tischler
Editor
Microbiology Spectrum

Reviewer #3 (Comments for the Author):

Dear Authors,
This manuscript describes a new species of *Delftia* bacteria whose genome includes genes for aromatic compound metabolism pathways and the enzymes involved in these processes. The publication is well written, scientifically novel, and presents

interesting data.

I have a few comments:

In the Abstract:

The sentence in lines 38-41 should be revised either by listing all three values to which isDDH, ANI, and AAI are equal, or by clearly stating that isDDH is 70 % and that ANI and AAI reach 95-96 %. The abbreviation 'in silico DDH' is only shortened to isDDH in the abstract; elsewhere, it appears as 'in silico DDH' and 'DDH'. These abbreviations should be standardized. I recommend not using abbreviations in the abstract but writing them out once in the main text and then using this abbreviation throughout the rest of the paper. The same applies to other abbreviations, such as 3-HBA, 4-HBA, etc.

In line 44-48, it should be clarified that the activities are measured in cell-free extracts. It may also be unclear to the reader why two specific activities are shown for gentisate 1,2-dioxygenase. The sentence could be rewritten as follows: 'We determined the specific activities of key metabolic enzymes in cell-free extracts, including catechol-1,2-dioxygenase, catechol-2,3-dioxygenase, protocatechuic acid 4,5-dioxygenase and gentisate 1,2-dioxygenase.'

48 - 50 line: „Growth studies and enzymatic activity analysis revealed that strain PS-11 possesses functional metabolic pathways to metabolize various aromatic compounds as the sole source of carbon and energy". This sentence is unnecessary; I recommend deleting it.

Table 1.

The text mentions ANI, while the table shows the ortho ANI value. This should be unified.

Table 3.

Why was the activity of protocatechuate 3,4-dioxygenase not measured based on product formation, as in the case of protocatechuate 4,5-dioxygenase? Table 3 shows the activity of protocatechuate 4,5-dioxygenase but no activity of protocatechuate 3,4-dioxygenase. The substrate for both enzymes is the same. How would you explain this discrepancy, given that if protocatechuate 4,5-dioxygenase activity is observed in an extract grown with 4-hydroxybenzoic acid, its product consumption should also be observed when measuring protocatechuate 3,4-dioxygenase activity?

This discrepancy should be clarified.

This manuscript describes a new species of *Delftia* bacteria whose genome includes genes for aromatic compound metabolism pathways and the enzymes involved in these processes. The publication is well written, scientifically novel, and presents interesting data.

I have a few comments:

In the Abstract:

The sentence in lines 38-41 should be revised either by listing all three values to which isDDH, ANI, and AAI are equal, or by clearly stating that isDDH is 70 % and that ANI and AAI reach 95-96 %. The abbreviation in 'in silico DDH' is only shortened to isDDH in the abstract; elsewhere, it appears as 'in silico DDH' and 'DDH'. These abbreviations should be standardized. I recommend not using abbreviations in the abstract but writing them out once in the main text and then using this abbreviation throughout the rest of the paper. The same applies to other abbreviations, such as 3-HBA, 4-HBA, etc.

In line 44-48, it should be clarified that the activities are measured in cell-free extracts. It may also be unclear to the reader why two specific activities are shown for gentisate 1,2-dioxygenase. The sentence could be rewritten as follows: 'We determined the specific activities of key metabolic enzymes in cell-free extracts, including catechol-1,2-dioxygenase, catechol-2,3-dioxygenase, protocatechuic acid 4,5-dioxygenase and gentisate 1,2-dioxygenase.'

48 – 50 line: „Growth studies and enzymatic activity analysis revealed that strain PS-11 possesses functional metabolic pathways to metabolize various aromatic compounds as the sole source of carbon and energy“. This sentence is unnecessary; I recommend deleting it.

Table 1.

The text mentions ANI, while the table shows the ortho ANI value. This should be unified.

Table 3.

Why was the activity of protocatechuate 3,4-dioxygenase not measured based on product formation, as in the case of protocatechuate 4,5-dioxygenase? Table 3 shows the activity of protocatechuate 4,5-dioxygenase but no activity of protocatechuate 3,4-dioxygenase. The substrate for both enzymes is the same. How would you explain this discrepancy, given that if protocatechuate 4,5-dioxygenase activity is observed in an extract grown with 4-hydroxybenzoic acid, its product consumption should also be observed when measuring protocatechuate 3,4-dioxygenase activity? This discrepancy should be clarified.

Response to the reviewer's comments

We are grateful to the editor and reviewers for reviewing our manuscript and providing us necessary comments and suggestions for improving our manuscript. Please find below Response to the reviewer's comments for your kind consideration

In the Abstract:

The sentence in lines 38-41 should be revised either by listing all three values to which isDDH, ANI, and AAI are equal, or by clearly stating that isDDH is 70 % and that ANI and AAI reach 95-96 %. The abbreviation in 'in silico DDH is only shortened to isDDH in the abstract; elsewhere, it appears as 'in silico DDH' and 'DDH'. These abbreviations should be standardized. I recommend not using abbreviations in the abstract but writing them out once in the main text and then using this abbreviation throughout the rest of the paper. The same applies to other abbreviations, such as 3-HBA, 4-HBA, etc.

#Reply: We appreciate the reviewer's comments. As suggested, necessary corrections has been done in the revised manuscript and in the Table 1.

In line 44-48, it should be clarified that the activities are measured in cell-free extracts. It may also be unclear to the reader why two specific activities are shown for gentisate 1,2-dioxygenase. The sentence could be rewritten as follows: 'We determined the specific activities of key metabolic enzymes in cell-free extracts, including catechol-1,2-dioxygenase, catechol-2,3-dioxygenase, protocatechuic acid 4,5-dioxygenase and gentisate 1,2-dioxygenase.'

#Reply: We appreciate the reviewer's comments. As suggested, we have rewritten in the line 44-48 in the revised manuscript.

48 - 50 line: „Growth studies and enzymatic activity analysis revealed that strain PS-11 possesses functional metabolic pathways to metabolize various aromatic compounds as the sole source of carbon and energy". This sentence is unnecessary; I recommend deleting it.

#Reply: We appreciate the reviewer's comments. As suggested, we have deleted the above lines in the revised manuscript.

Table 1. The text mentions ANI, while the table shows the ortho ANI value. This should be unified.

#Reply: We appreciate the reviewer's comments. As suggested, necessary corrections has been done in the revised manuscript and in the Table 1.

Table 3.

Why was the activity of protocatechuate 3,4-dioxygenase not measured based on product formation, as in the case of protocatechuate 4,5-dioxygenase? Table 3 shows the activity of protocatechuate 4,5-dioxygenase but no activity of protocatechuate 3,4-dioxygenase. The substrate for both enzymes is the same. How would you explain this discrepancy, given that if protocatechuate 4,5-dioxygenase activity is observed in an extract grown with 4-hydroxybenzoic acid, its product consumption should also be observed when measuring

protocatechuate 3,4-dioxygenase activity?
This discrepancy should be clarified.

#Reply: We appreciate the reviewer's comments. To decipher the mode of ring-cleaving protocatechuate dioxygenase in this strain, time-dependent spectral changes were monitored using cell-free extracts with 3,4-dihydroxybenzoate (protocatechuate) as the substrate. Protocatechuate can undergo ring cleavage *via* two possible pathways: (i) *ortho*-ring cleavage, catalyzed by protocatechuate 3,4-dioxygenase (P34DO), producing a colorless product, β -carboxy-*cis,cis*-muconic acid (λ_{\max} at 260 nm); and (ii) *meta*-ring cleavage, catalyzed by protocatechuate 4,5-dioxygenase (P45DO), producing α -hydroxy- γ -carboxymuconic semialdehyde, a yellow-colored product with a λ_{\max} at 410 nm.

It is error-prone to measure the activity of P34DO in cell-free extracts (CFE) by monitoring product formation at 260 nm due to interference from substrate as well as other aromatic metabolites present in the CFE. Therefore, time-dependent spectral changes were recorded in the presence of protocatechuate and the CFE of strain PS-11 grown on 4-hydroxybenzoic acid. The spectra showed a decrease in the absorbance at 290 nm (λ_{\max} of protocatechuate) along with a concomitant increase in the absorbance at 410 nm, indicating the activity of P45DO. The absence of characteristic spectral features corresponding to the *ortho*-cleavage product, together with the absence of the gene encoding P34DO in strain PS-11, confirmed the absence of the P34DO enzyme. Thus, the combined genomic and enzymatic analyses indicate the presence of P45DO and the absence of P34DO in this strain. This discrepancy has now been clarified in the revised manuscript.

Re: Spectrum03105-25R2 (Genomic and functional insights into aromatic compound degradation by *Delftia* strain PS-11 isolated from freshwater pufferfish)

Dear Prof. Subrata Das:

Your manuscript has been accepted, and I am forwarding it to the ASM production staff for publication. Your paper will first be checked to make sure all elements meet the technical requirements. ASM staff will contact you if anything needs to be revised before copyediting and production can begin. Otherwise, you will be notified when your proofs are ready to be viewed.

Sincerely,
Dirk Tischler
Editor
Microbiology Spectrum